# Weakly Supervised 3D Open-vocabulary Segmentation

**Kunhao Liu**[1]   **Fangneng Zhan**[2]   **Jiahui Zhang**[1]   **Muyu Xu**[1]   **Yingchen Yu**[1]
**Abdulmotaleb El Saddik**[3,5]   **Christian Theobalt**[2]   **Eric Xing**[4,5]   **Shijian Lu**[1*]

[1]Nanyang Technological University   [2]Max Planck Institute for Informatics
[3]University of Ottawa   [4]Carnegie Mellon University   [5]MBZUAI

## Abstract

Open-vocabulary segmentation of 3D scenes is a fundamental function of human perception and thus a crucial objective in computer vision research. However, this task is heavily impeded by the lack of large-scale and diverse 3D open-vocabulary segmentation datasets for training robust and generalizable models. Distilling knowledge from pre-trained 2D open-vocabulary segmentation models helps but it compromises the open-vocabulary feature as the 2D models are mostly finetuned with close-vocabulary datasets. We tackle the challenges in 3D open-vocabulary segmentation by exploiting pre-trained foundation models CLIP and DINO in a weakly supervised manner. Specifically, given only the open-vocabulary text descriptions of the objects in a scene, we distill the open-vocabulary multimodal knowledge and object reasoning capability of CLIP and DINO into a neural radiance field (NeRF), which effectively lifts 2D features into view-consistent 3D segmentation. A notable aspect of our approach is that it does not require any manual segmentation annotations for either the foundation models or the distillation process. Extensive experiments show that our method even outperforms fully supervised models trained with segmentation annotations in certain scenes, suggesting that 3D open-vocabulary segmentation can be effectively learned from 2D images and text-image pairs. Code is available at `https://github.com/Kunhao-Liu/3D-OVS`.

## 1   Introduction

Semantic segmentation of 3D scenes holds significant research value due to its broad range of applications such as robot navigation [1], object localization [2], autonomous driving [3], 3D scene editing [4], augmented/virtual reality, etc. Given the super-rich semantics in 3D scenes, a crucial aspect of this task is achieving open-vocabulary segmentation that can handle regions and objects of various semantics including those with long-tail distributions. This is a grand challenge as it necessitates a comprehensive understanding of natural language and the corresponding objects in the 3D world.

The main challenge in open-vocabulary 3D scene segmentation is the lack of large-scale and diverse 3D segmentation datasets. Existing 3D segmentation datasets like ScanNet [5] primarily focus on restricted scenes with limited object classes, making them unsuitable for training open-vocabulary models. An alternative is to distill knowledge from pre-trained 2D open-vocabulary segmentation models to 3D representations as learned with NeRF [6] or point clouds, by fitting the feature maps or segmentation probability outputs from the 2D models [4, 7]. Though this approach circumvents the need for the 3D datasets, it inherits the limitations of the 2D models which are usually finetuned with close-vocabulary datasets of limited text labels [8, 9], thereby compromising the open-vocabulary property, especially for text labels with long-tail distributions [2, 3].

---

[*]Corresponding author

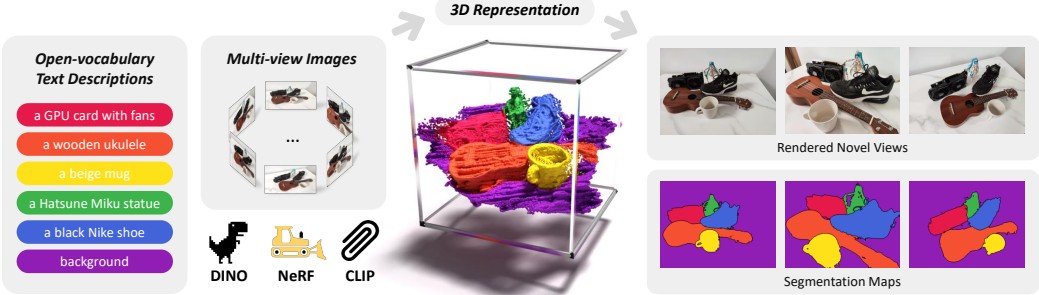

Figure 1: **Weakly Supervised 3D Open-vocabulary Segmentation.** Given the multi-view images of a 3D scene and the open-vocabulary text descriptions, our method distills open-vocabulary multimodal knowledge from CLIP and object reasoning ability from DINO into the reconstructed NeRF, producing accurate object boundaries for the 3D scene without requiring any segmentation annotations during training.

We achieve precise and annotation-free 3D open-vocabulary segmentation by distilling knowledge from two pre-trained foundation models into NeRF in a weakly supervised manner, supervised only by the open-vocabulary text descriptions of the objects in a scene, as illustrated in Fig. 1. One foundation model is CLIP [10] which is trained with Internet-scale text-image pairs [11] capturing extensive open-vocabulary multimodal knowledge. The other is DINO [12, 13] which is trained with large-scale unlabelled images capturing superb scene layout and object boundary information. However, CLIP yields image-level features which are not suitable for pixel-level semantic segmentation. Thus certain mechanisms should be designed to extract pixel-level CLIP features without fine-tuning. Additionally, the image patches' CLIP features may have ambiguities for segmentation, which need to be regularized for accurate open-vocabulary segmentation. At the other end, DINO produces feature maps instead of explicit segmentation maps. Certain distillation techniques should be designed to extract the necessary information from DINO features to facilitate precise segmentation.

We construct a hierarchical set of image patches to extract pixel-level features from image-level CLIP features and design a 3D *Selection Volume* to identify the appropriate hierarchical level for each 3D point, effectively aligning CLIP features with pixel-level features without fine-tuning. In addition, we introduce a *Relevancy-Distribution Alignment (RDA)* loss to address CLIP feature ambiguities, aligning segmentation probability distribution with class relevancies that capture similarities between class text features and corresponding CLIP features. Moreover, we propose a novel *Feature-Distribution Alignment (FDA)* loss to distill object boundary information from DINO features. The FDA loss encourages close segmentation probability distributions for points with similar DINO features and distant distributions for dissimilar features. To address the training instability due to diverse distribution shapes, we further re-balance weights associated with similar and dissimilar DINO features.

Our method enables weakly supervised open-vocabulary segmentation of 3D scenes with accurate object boundaries. By distilling knowledge from CLIP without fine-tuning, our approach preserves its open-vocabulary knowledge and effectively handles text labels with long-tail distributions. A notable aspect of our approach is that it does not require any manual segmentation annotations for either the foundation models or the distillation process. Remarkably, our experiments demonstrate that our method surpasses fully supervised models trained with segmentation annotations in certain scenes, highlighting the possibility that 3D open-vocabulary segmentation can be effectively learned from large amounts of 2D images and text-image pairs.

In summary, the contributions of this work are three-fold. *Firstly*, we propose an innovative pipeline for weakly supervised 3D open-vocabulary segmentation by distilling knowledge from pre-trained foundation models into NeRF without requiring any annotations in training. *Secondly*, we introduce a Selection Volume to align image-level CLIP features with pixel-level features, supplemented by novel Relevancy-Distribution Alignment and Feature-Distribution Alignment losses that respectively resolve CLIP features' ambiguities and effectively distill DINO features for 3D scene segmentation. *Lastly*, extensive experiments demonstrate that our method effectively recognizes long-tail classes and produces accurate segmentation maps, even with limited input data.

## 2   Related Work

**Open-vocabulary Segmentation.**   In recent years, the field of 2D open-vocabulary segmentation has garnered significant attention, driven by the availability of extensive text-image datasets and vast computational resources. Predominant approaches [8, 14–19] typically distill knowledge from large-scale pre-trained models, such as image-text contrastive learning models [10, 20–22] and diffusion models [23]. However, the distillation process requires fine-tuning on close-vocabulary datasets, contrasting with massive datasets used for large-scale pre-trained models [11]. This leads to limited performance in recalling infrequent classes with long-tail distributions [2, 3], compromising the open-vocabulary property. OpenSeg [24] is not finetuned on a closed set of classes but is weakly supervised via image captions. However, OpenSeg has a smaller vocabulary and knowledge than CLIP as it is trained on a much smaller dataset. Our method, without fine-tuning CLIP, effectively handles such classes.

**3D Scenes Segmentation.**   3D scene segmentation has been a long-standing challenge in computer vision. Traditional approaches focus on point clouds or voxels with limited class variety in datasets, restricting generalizability to unseen classes [5, 25–36]. Recently, numerous point-cloud-based techniques have emerged to explore open-vocabulary 3D scene segmentation by encoding 2D open-vocabulary models' features into 3D scene points [3, 7, 37–40]. However, these methods are also mostly evaluated on datasets with restricted scenes and limited class ranges [5, 27, 28, 41], not fully exhibiting the open-vocabulary property. Moreover, point clouds have compromised geometric details, making them less suitable for precise segmentation compared to NeRF representations [6, 42]. Consequently, there has been a surge in NeRF-based 3D segmentation techniques that mainly address interactive segmentation [4, 43, 44], panoptic segmentation [45, 46], moving part segmentation [47], object part segmentation [48], object co-segmentation [49], unsupervised object segmentation [50, 51], etc. FFD [4] attempts to segment unseen text labels during training by fitting LSeg's [8] feature maps to a NeRF, but inherits LSeg's limitations, hindering generalization to long-tail distribution classes. Our method overcomes these challenges by directly using CLIP image features and distilling them into a NeRF representation [42] without fine-tuning on close-vocabulary datasets.

**Foundation Models.**   Pre-trained foundation models [52, 53] have become a powerful paradigm in computer science due to their ability to capture general knowledge and adapt to various downstream tasks [10, 12, 13, 20, 23, 54–57]. These models are trained using various paradigms in natural language processing, such as masked language modeling [57, 58], denoising autoencoder [59], replaced token detection [60], and sentence prediction tasks[61], as well as in computer vision, including data generation [23, 62, 63], data reconstruction [64], and data contrastive learning [10, 12, 13, 20–22]. Foundation models acquire emergent capabilities for exceptional performance on downstream tasks, either in a zero-shot manner or with fine-tuning. In this work, we harness the capabilities of two prominent foundation models, CLIP [10] and DINO [12, 13]. CLIP learns associations between images and texts by mapping them to a shared space, facilitating applications in tasks like image classification, object detection, visual question-answering, and image generation [9, 10, 23, 62, 65, 66]. DINO, trained in a self-supervised manner, extracts scene layout information, particularly object boundaries, and has been successfully employed in tasks such as classification, detection, segmentation, keypoint estimation, depth estimation, and image editing [12, 13, 49, 67–69].

## 3   Method

We propose a novel method for weakly supervised open-vocabulary segmentation of reconstructed NeRF. Given the multi-view images of a scene and the open-vocabulary text description for each class, we aim to segment the reconstructed NeRF such that every 3D point is assigned a corresponding class label.

To achieve this, we exploit the CLIP model's multimodal knowledge by mapping each 3D point to a CLIP feature representing its semantic meaning. As CLIP only generates image-level features, we extract a hierarchy of CLIP features from image patches and learn a 3D *Selection Volume* for pixel-level feature extraction, as described in Sec. 3.1.

---

**Algorithm 1:** Extracting pixel-level features of an image from CLIP

---

**Input:** RGB image $I \in \mathbb{R}^{3 \times H \times W}$, number of scale $N_s$, CLIP image encoder $E_i$

**Output:** multi-scale pixel-level features $F_I \in \mathbb{R}^{N_s \times D \times H \times W}$

**Initialize:** the patch size of each scale $patch\_sizes \in \mathbb{R}^{N_s}$, $F_I = \text{zeros}(N_s, D, H, W)$

/* Loop over all the scales */
**for** $scale\_idx, patch\_size$ **in** enumerate($patch\_sizes$) **do**
   $stride = patch\_size/4$
   $count = \text{zeros}(1,1,H,W)$ /* Record the patch count for each pixel */
   $F_{multi\_spatial} = \text{zeros}(1,D,H,W)$ /* Multi-spatial feature of current scale */
   /* Loop over all the patches */
   **for** $x\_idx$ **in** range($(H - patch\_size)/stride + 1$) **do**
      $start\_x = x\_idx \times stride$
      **for** $y\_idx$ **in** range($(W - patch\_size)/stride + 1$) **do**
         $start\_y = y\_idx \times stride$
         /* Get image patch's coordinates with randomness */
         $(left, upper, right, lower) = (\max(start\_y - \text{randint}(0, stride), 0), \max(start\_x - \text{randint}(0, stride), 0),$
            $\min(start\_y + patch\_size + \text{randint}(0, stride), W), \min(start\_x + patch\_size + \text{randint}(0, stride), H))$
         /* Get image patch's CLIP feature */
         $F_{patch} = E_i(I.\text{crop}(left, upper, right, lower))$
         $F_{multi\_spatial}[:, :, upper:lower, left:right] += F_{patch}$
         $count[:, :, upper:lower, left:right] += 1$
      **end**
   **end**
   $F_{multi\_spatial} /= count$
   $F_I[scale\_idx] = F_{multi\_spatial}$
**end**

---

However, the image patches' CLIP features may have ambiguities for segmentation, showing inaccurate absolute relevancy values which lead to misclassification. Take an image patch capturing an apple lying on a lawn as an example. The corresponding CLIP feature contains both apple and lawn information. If the apple is relatively small, the patch could be classified as lawn because lawn dominates the patch's CLIP features. Then the class apple would be ignored. To address this issue, we introduce a *Relevancy-Distribution Alignment (RDA)* loss, aligning the segmentation probability distribution with each class's normalized relevancy map, as described in Sec. 3.2. For precise object boundaries, we align the segmentation probability distribution with the images' DINO features. Previous segmentation approaches utilizing DINO features have focused on unsupervised segmentation [49, 68], lacking semantic meaning for segmented parts. In the open-vocabulary context, assigning accurate text labels to segmented regions requires aligning DINO feature clusters with correct text labels. To overcome this challenge, we propose a *Feature-Distribution Alignment (FDA)* loss, segmenting the scene based on DINO features' distribution and assigning appropriate text labels, as described in Sec. 3.3.

## 3.1 Distilling Pixel-level CLIP Features with a 3D Selection Volume

We propose a method based on multi-scale and multi-spatial strategies to adapt CLIP's image-level features for pixel-level segmentation, motivated by the observation that a pixel's semantic meaning should remain invariant to its surrounding pixels. The multi-scale component extracts features from patches of varying sizes around each pixel, and the multi-spatial component extracts from patches in which each pixel is at different positions. We average the multi-spatial features in each scale, attributing the primary direction of these features to the pixel's semantic meaning. We utilize a sliding-window algorithm for multi-spatial feature extraction. To prevent checkerboard patterns in the features and potential segmentation artifacts, we introduce randomness in the window size. The algorithm's pseudo-code is in Alg. 1.

After extracting pixel-level features from CLIP, we now have the multi-view RGB images and their corresponding multi-scale pixel-level features. Each ray $\mathbf{r}$ is assigned its multi-scale feature $F(\mathbf{r}) \in \mathbb{R}^{N_s \times D}$ for supervision. Rather than simply averaging each ray's multi-scale features across scales, we introduce a 3D Selection Volume $S$ to determine the most suitable scale indicative of the object size within a patch. Each 3D point $\mathbf{x} \in \mathbb{R}^3$ yields a selection vector $S_{\mathbf{x}} \in \mathbb{R}^{N_s}$ from $S$. Following [4, 45], we introduce an additional branch to render the CLIP feature. We can then render the RGB value, the CLIP feature, and the selection vector of each ray $\mathbf{r}$ using volume rendering [6]:

$$\hat{C}(\mathbf{r}) = \sum_i T_i \alpha_i C_i \in \mathbb{R}^3, \quad \hat{F}(\mathbf{r}) = \sum_i T_i \alpha_i F_i \in \mathbb{R}^D, \tag{1}$$

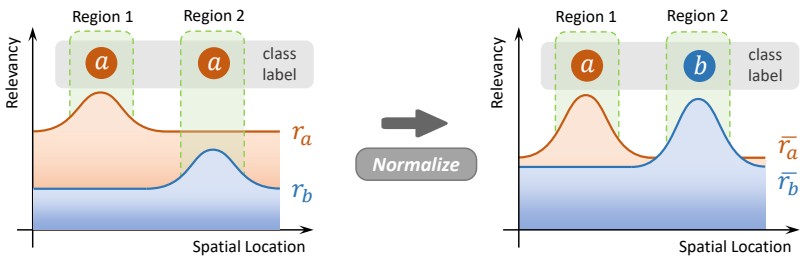

Figure 2: **Mitigating CLIP features' ambiguities with normalized relevancy maps.** For original relevancy maps $r_a, r_b$ of classes $a$ and $b$, we note a higher relevancy for class $b$ in Region 2 than in other image regions. Despite this, the ambiguities of CLIP features lead to Region 2's classification as $a$ due to the higher absolute relevancy of $a$ in Region 2, even as $a$ is located in Region 1. To rectify this, we normalize each class's relevancy maps to a fixed range. These normalized relevancy maps, $\bar{r}_a$ and $\bar{r}_b$, reduce such ambiguities, facilitating accurate region-class assignments.

$$S(\mathbf{r}) = \text{Softmax}\left(\sum_i T_i \alpha_i S_i\right) \in [0,1]^{N_s}, \tag{2}$$

where $C_i, F_i, S_i$ are the color, feature, and selection vector of each sampled point along the ray, $T_i = \Pi_{j=0}^{i-1}(1 - \alpha_i)$ is the accumulated transmittance and $\alpha_i = 1 - \exp(-\delta_i \sigma_i)$ is the opacity of the point. We apply a Softmax function to the selection vector of each ray such that the sum of the probability of each scale is equal to 1.

For a set of rays $\mathcal{R}$ in each training batch, the supervision loss can then be formulated as the combination of the L2 distance between rendered and ground truth RGB values and the cosine similarities $\cos\langle,\rangle$ between the rendered features and the selected multi-scale CLIP features:

$$\mathcal{L}_{supervision} = \sum_{\mathbf{r} \in \mathcal{R}} \left( \left\|\hat{C}(\mathbf{r}) - C(\mathbf{r})\right\|_2^2 - \cos\langle \hat{F}(\mathbf{r}), S(\mathbf{r})F(\mathbf{r})\rangle \right). \tag{3}$$

Given a set of text descriptions $\{[\text{CLASS}]_i\}_{i=1}^C$ of $C$ classes and the CLIP text encoder $E_t$, we can get the classes' text features $T = E_t([\text{CLASS}]) \in \mathbb{R}^{C \times D}$. Then we can get the segmentation logits $z(\mathbf{r})$ of the ray $\mathbf{r}$ by computing the cosine similarities between the rendered CLIP feature and the classes' text features:

$$z(\mathbf{r}) = \cos\langle T, \hat{F}(\mathbf{r})\rangle \in \mathbb{R}^C. \tag{4}$$

We can then get the class label of the ray $l(\mathbf{r}) = \text{argmax}(z(\mathbf{r}))$.

### 3.2 Relevancy-Distribution Alignment for Ambiguity Mitigation

To mitigate the ambiguities of the CLIP features, we propose to align the segmentation probability distribution with the spatially normalized relevancy maps of each class, enabling our method to identify specific image regions described by each class text, as illustrated in Fig. 2. The segmentation probability of each ray $P(\mathbf{r})$ can be derived from the segmentation logits with a Softmax function:

$$P(\mathbf{r}) = \text{Softmax}(z(\mathbf{r})) \in [0,1]^C. \tag{5}$$

The relevancy of a given class is determined by the similarity between the class's text feature and the selected feature from the hierarchy of image patches' CLIP features. Given an image $I$, we can get its multi-scale pixel-level CLIP feature $F_I \in \mathbb{R}^{N_s \times D \times H \times W}$ using Alg. 1 and selection vector $S_I \in \mathbb{R}^{N_s \times H \times W}$ using Eq. (2). And then we can get the image's relevancy map $R_I \in \mathbb{R}^{C \times H \times W}$ as:

$$R_{I_{hw}} = S_{I_{hw}} \cos\langle T, F_{I_{hw}}\rangle, \tag{6}$$

where where $h, w$ denotes the index in the $H$ and $W$ channel. We normalize each class's relevancy independently within an input view to $[0,1]$ to mitigate the ambiguities of CLIP features, making our method discern image regions described by each class text:

$$\bar{R}_I = (R_I - \min(R_I)) / (\max(R_I) - \min(R_I)) \in [0,1]^{C \times H \times W}, \tag{7}$$

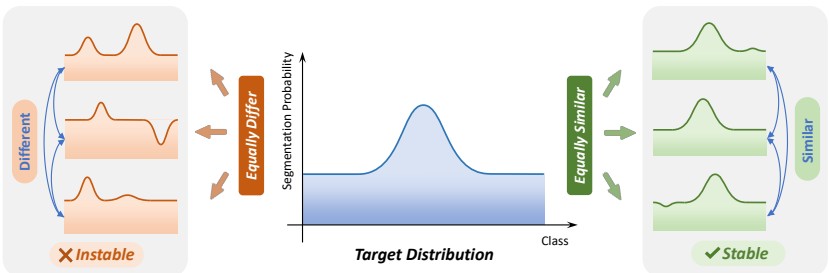

Figure 3: **Difference between similar and distant distributions.** Distributions having large divergence from the target distribution exhibit significantly diverse shapes, increasing the training instability (left). Conversely, distributions displaying low divergence with the target distribution consistently demonstrate a similar shape (right).

where $\min()$ and $\max()$ are the functions getting the lowest and highest values across the spatial dimensions (i.e. $H$ and $W$). We apply a Softmax function to $\bar{R}_I$ to make it a probability vector. Then we can assign each ray $\mathbf{r}$ its normalized relevancy with all the classes $\bar{R}(\mathbf{r}) \in [0,1]^C$. We employ the Jensen-Shannon (JS) divergence to measure the discrepancy between the normalized relevancy $\bar{R}(\mathbf{r})$ and the segmentation probability distribution $P(\mathbf{r})$ of each ray, formulating the Relevancy-Distribution Alignment (RDA) loss:

$$\mathcal{L}_{RDA} = \sum_{\mathbf{r} \in \mathcal{R}} \sum_{c \in C} \left( P(\mathbf{r})_c \log \left( \frac{P(\mathbf{r})_c}{M_{P\bar{R}}(\mathbf{r})_c} \right) + \bar{R}(\mathbf{r})_c \log \left( \frac{\bar{R}(\mathbf{r})_c}{M_{P\bar{R}}(\mathbf{r})_c} \right) \right) / 2, \tag{8}$$

where $M_{P\bar{R}}(\mathbf{r}) = (P(\mathbf{r}) + \bar{R}(\mathbf{r}))/2$ is the average of the two distributions, and the subscript $c$ denotes the probability of the $c$th class. By aligning the normalized relevancies and the segmentation probability distributions, our method can effectively identify the specific region corresponding to the text description of each class.

### 3.3 Feature-Distribution Alignment for Precise Object Boundary Segmentation

To ensure the segmentation exhibits precise object boundaries, we align the segmentation probability distribution with the images' DINO features, which have been shown to capture superb scene layouts and object boundary information [12, 13]. Following [49, 68], we extract the scene layout information with a DINO feature correlation tensor. Given a patch of size $H_p \times W_p$, we can get the correlation tensor $Corr\_F \in \mathbb{R}^{H_p W_p \times H_p W_p}$ as:

$$Corr\_F_{hwij} = \cos\langle f_{hw}, f_{ij} \rangle, \tag{9}$$

whose entries represent the cosine similarity between the DINO features $f$ at spatial positions $(h, w)$ and $(i, j)$ of the patch. In order to construct the correlation tensor for the segmentation probability distribution, we propose utilizing the JS divergence to assess the similarity between segmentation probabilities at two distinct spatial positions. The choice of JS divergence offers several advantages, including its symmetric nature and a bounded range of $[0, 1]$, which contribute to improved numerical stability. However, since we only care about the class label of each point, i.e. the entry with the highest probability, we use a low temperature $\tau < 1$ to get a sharper version of the segmentation probability distribution $\acute{P}$ to let the model focus on the entry with the largest probability:

$$\acute{P} = \text{Softmax}\left(z/\tau\right) \in [0, 1]^C. \tag{10}$$

The distribution correlation tensor $Corr\_D \in \mathbb{R}^{H_p W_p \times H_p W_p}$ can thus be computed with:

$$Corr\_D_{hwij} = \sum_{c \in C} \left( \acute{P}_{hwc} \log \left( \frac{\acute{P}_{hwc}}{M_{\acute{P}\acute{P}c}} \right) + \acute{P}_{ijc} \log \left( \frac{\acute{P}_{ijc}}{M_{\acute{P}\acute{P}c}} \right) \right) / 2, \tag{11}$$

where $\acute{P}_{hwc}, \acute{P}_{ijc}$ are the segmentation probabilities of the $c$th class at spatial locations $(h, w)$ and $(i, j)$ of the patch, $M_{\acute{P}\acute{P}} = (\acute{P}_{hw} + \acute{P}_{ij})/2$ is the average of the two distributions. Thus the correlation loss [68] can be expressed as:

$$\mathcal{L}_{corr} = \sum_{hwij} (Corr\_F_{hwij} - b) \times Corr\_D_{hwij}, \tag{12}$$

Table 1: **Quantitative comparisons.** We report the mIoU(↑) scores and the Accuracy(↑) scores of the following methods in 6 scenes and highlight the best , second-best , and third-best scores. Our method outperforms both 2D and 3D methods without any segmentation annotations in training.

| | Methods | bed | | sofa | | lawn | | room | | bench | | table | |
|---|---|---|---|---|---|---|---|---|---|---|---|---|---|
| | | mIoU | Accuracy | mIoU | Accuracy | mIoU | Accuracy | mIoU | Accuracy | mIoU | Accuracy | mIoU | Accuracy |
| 2D | LSeg [8] | 56.0 | 87.6 | 04.5 | 16.5 | 17.5 | 77.5 | 19.2 | 46.1 | 06.0 | 42.7 | 07.6 | 29.9 |
| | ODISE [16] | 52.6 | 86.5 | 48.3 | 35.4 | 39.8 | 82.5 | 52.5 | 59.7 | 24.1 | 39.0 | 39.7 | 34.5 |
| | OV-Seg [19] | 79.8 | 40.4 | 66.1 | 69.6 | 81.2 | 92.1 | 71.4 | 49.1 | 88.9 | 89.2 | 80.6 | 65.3 |
| 3D | FFD [4] | 56.6 | 86.9 | 03.7 | 09.5 | 42.9 | 82.6 | 25.1 | 51.4 | 06.1 | 42.8 | 07.9 | 30.1 |
| | Sem(ODISE) [45] | 50.3 | 86.5 | 27.7 | 22.2 | 24.2 | 80.5 | 29.5 | 61.5 | 25.6 | 56.4 | 18.4 | 30.8 |
| | Sem(OV-Seg) [45] | 89.3 | 96.7 | 66.3 | 89.0 | 87.6 | 95.4 | 53.8 | 81.9 | 94.2 | 98.5 | 83.8 | 94.6 |
| | LERF [2] | 73.5 | 86.9 | 27.0 | 43.8 | 73.7 | 93.5 | 46.6 | 79.8 | 53.2 | 79.7 | 33.4 | 41.0 |
| | **Ours** | 89.5 | 96.7 | 74.0 | 91.6 | 88.2 | 97.3 | 92.8 | 98.9 | 89.3 | 96.3 | 88.8 | 96.5 |

where $b$ is a hyper-parameter denoting that we consider the segmentation probabilities of two spatial locations $(h, w)$ and $(i, j)$ to be similar if their DINO features' similarity is larger than $b$ and distant if less than $b$. Nonetheless, the correlation loss $\mathcal{L}_{corr}$ introduces significant instability due to the diverse shapes of distributions with large divergence from a target distribution, making the loss assign wrong labels to the segmented parts. Conversely, when a distribution displays a low JS divergence with the target distribution, it consistently demonstrates a similar shape to the target distribution, as shown in Fig. 3. Based on this observation, we propose re-balancing the weights associated with similar and dissimilar DINO features. Specifically, we allocate a much greater weight to the correlation loss arising from similar DINO features and a smaller weight to that of dissimilar DINO features, thereby mitigating the instability caused by the correlation loss. Thus the Feature-Distribution Alignment (FDA) loss can be formulated with:

$$pos\_F = \text{clamp}(Corr\_F - b, \min = 0), \quad neg\_F = \text{clamp}(Corr\_F - b, \max = 0), \quad (13)$$

$$\mathcal{L}_{FDA} = \lambda_{pos} \sum_{hwij} (pos\_F_{hwij} \times Corr\_D_{hwij})/\text{count\_nonzero}(pos\_F)+$$

$$\lambda_{neg} \sum_{hwij} (neg\_F_{hwij} \times Corr\_D_{hwij})/\text{count\_nonzero}(neg\_F), \quad (14)$$

where $\text{clamp}(, \min/\max = 0)$ is to clamp all the elements smaller/greater than 0 to 0, thus making $pos\_F \geq 0$ and $neg\_F \leq 0$, $\text{count\_nonzero}()$ is to count the number of non zero elements, and $\lambda_{pos}, \lambda_{neg}$ are the weights associated with similar and dissimilar DINO features, which are set as $\lambda_{pos} = 200 \gg \lambda_{neg} = 0.2$ by default.

The total training loss is: $\mathcal{L} = \mathcal{L}_{supervision} + \mathcal{L}_{RDA} + \mathcal{L}_{FDA}$, where $\mathcal{L}_{supervision}, \mathcal{L}_{RDA}$ are calculated with randomly sampled rays and $\mathcal{L}_{FDA}$ is calculated with randomly sampled patches.

## 4 Experiments

We evaluate our method on 3D open-vocabulary segmentation, showing that our method can recognize long-tail classes and produce highly accurate object boundaries even with limited input data. We employ TensoRF [42] as the backbone and extract 3 scales of pixel-level CLIP features. More implementation details and experiments are in the appendix.

**Dataset.** Existing 3D segmentation datasets predominantly focus on either restricted scenes with a narrow range of object classes [5, 70], or individual objects [71–73], thereby limiting their capacity to fully assess the task of 3D open-vocabulary segmentation. Thus following [2], we create a dataset comprising 10 distinct scenes. Each scene features a set of long-tail objects situated in various poses and backgrounds. Ground truth masks for the test views are manually annotated, enabling both qualitative and quantitative evaluation of our segmentation methods. We also evaluate our method on more diverse datasets which include human body, human head, indoor scenes with low-quality images [70, 74], and a complex scene from LERF datasets [2], as shown in the appendix.

**Baselines.** We benchmark our method with three NeRF-based methods capable of 3D open-vocabulary segmentation: FFD [4], Semantic-NeRF (abbreviated as Sem) [45], and LERF [2].

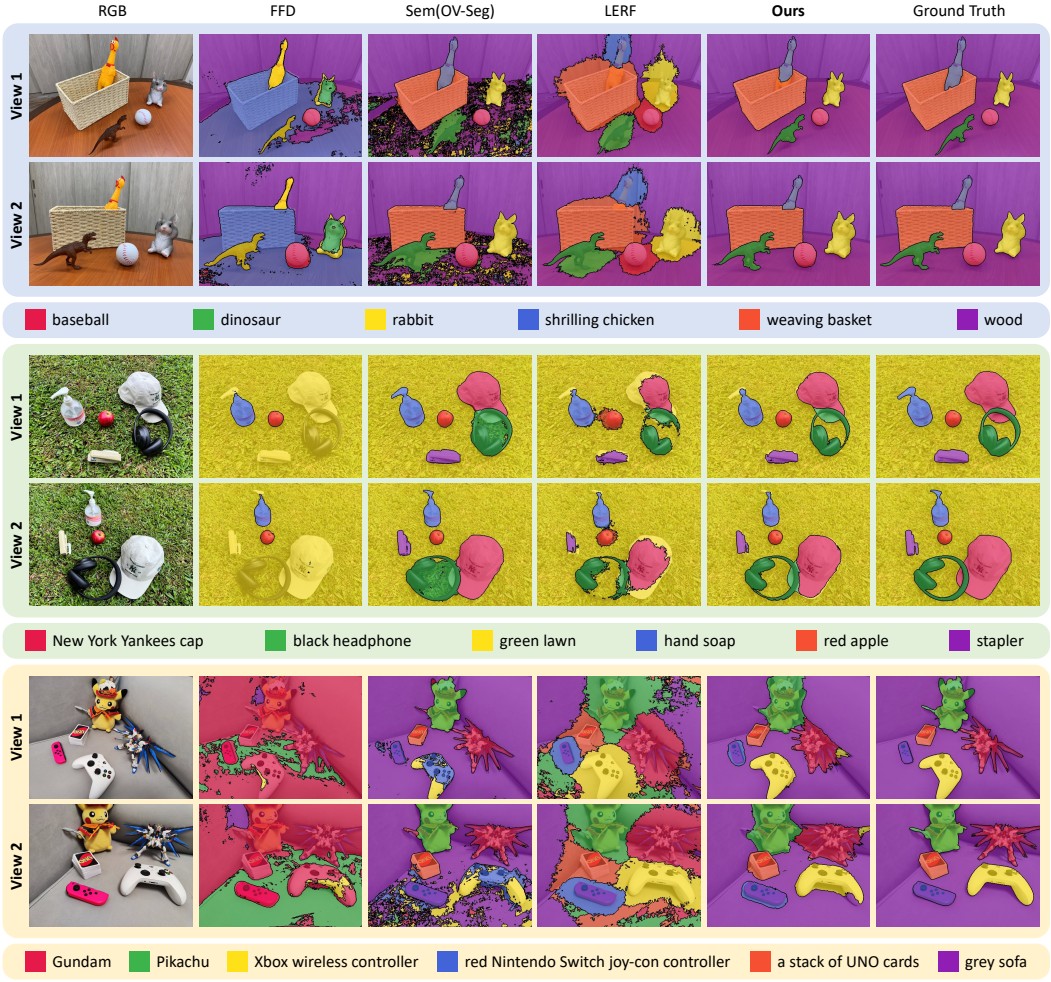

Figure 4: **Qualitative comparisons.** Visualization of the segmentation results in 3 scenes. Our method successfully recognizes long-tail classes and produces the most accurate segmentation maps.

FFD, a pioneering effort in 3D open-vocabulary segmentation, applies the feature maps of the 2D open-vocabulary segmentation method, LSeg [8], to achieve its segmentation results. For Sem, we adapt it to the open-vocabulary segmentation context by distilling segmentation results from two state-of-the-art 2D open-vocabulary segmentation models: the diffusion-model-based ODISE [16], and the CLIP-based OV-Seg [19]. LERF closely aligns with our proposed approach due to its use of knowledge distillation from CLIP and DINO. However, its primary focus is on object localization rather than segmentation. We use the same scale level number and patch sizes in LERF for fair comparisons. We also include results obtained by independently segmenting each test view using the aforementioned 2D models. Note that under our settings, FFD, Sem are fully supervised methods using segmentation annotations.

## 4.1 Results

We present the qualitative results in Fig. 4 and quantitative results in Tab. 1. Our proposed method outperforms all other techniques, including those which heavily rely on extensive segmentation annotations, such as LSeg, ODISE, OV-Seg. In particular, ODISE and FFD underperform in our evaluation, as they are unable to identify many long-tail classes, suggesting that the fine-tuning process of LSeg and ODISE may have led to a significant loss of the open-vocabulary knowledge originally encapsulated by CLIP and Stable Diffusion [23]. OV-Seg attempts to retain CLIP's knowledge by leveraging a mined dataset, however, it requires a mask proposal model which produces inconsistent segmentation across views, making Sem(OV-Seg) produce noisy and imprecise segmentation. LERF

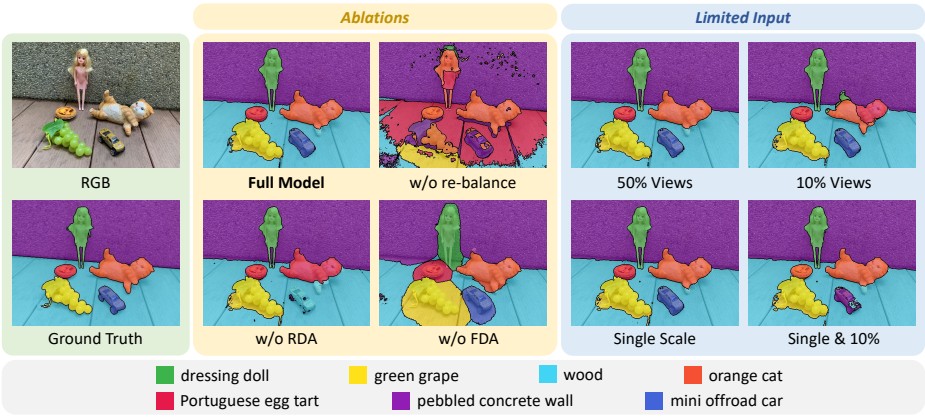

Figure 5: **Studies.** Visualization of the studies on ablations and limited input.

also fails to capture precise object boundaries due to its usage of a relatively naïve regularization loss, which fails to fully exploit the object boundary information within the DINO features. In contrast, our method exhibits robust performance, successfully recognizing long-tail classes and generating accurate and well-defined boundaries for each class. However, LERF allows querying any object without the need for running optimization again, which is an advantage over our method.

## 4.2 Studies

**Ablations.** We conduct ablation studies to evaluate the individual contributions of the RDA loss and the FDA loss to the overall performance of our proposed method. As shown in Tab. 2, both RDA loss and FDA loss are crucial to our method, without each of which can result in severe performance degradation. As illustrated in Fig. 5, without the RDA loss, the model does not resolve the ambiguities of the CLIP features, leading to misclassifications. For instance, it fails to distinguish between an *orange cat* and a *Portuguese egg tart*, and confuses a *mini offroad car* with *wood*. Without the FDA loss, although our method can correctly locate each class, it fails to segment precise object boundaries. When discarding the re-balancing in the FDA loss, i.e. using the correlation loss [68], the model produces accurate boundaries but assigns each cluster the wrong label due to the instability brought by diverse distribution shapes.

Table 2: **Studies.** We report the mean mIoU(↑) scores and the Accuracy(↑) scores in our studies.

|  | mIoU | Accuracy |
|---|---|---|
| w/o RDA loss | 57.2 | 79.4 |
| w/o FDA loss | 58.2 | 82.7 |
| w/o re-balance | 44.9 | 74.3 |
| 50% views | 85.7 | 95.7 |
| 10% views | 79.1 | 94.6 |
| single scale | 85.2 | 95.5 |
| single & 10% | 77.1 | 94.6 |
| **full model** | 86.2 | 95.8 |

**Limited input.** Given the substantial computational and storage demands of extracting hierarchical CLIP features for each view (exceeding 1GB for 3 scales in the full model), we explore whether reducing input CLIP features would yield similar results, as shown in Tab. 2 and Fig. 5. We test two modifications: reducing views for feature extraction and using a single scale of CLIP features rather than three. Halving the input views for feature extraction leads to negligible performance degradation ($< 1\%$) and minor visual differences compared to the full model. When reducing to only $10\%$ of input views, equivalent to 2-3 views in our dataset, we observe a modest $9\%$ drop in the mIoU score and a $1\%$ decrease in the Accuracy score, while retaining accurate segmentation across most classes. Using a single scale of CLIP features also only incurs minimal degradation ($< 1\%$). Even under extreme conditions, i.e., extracting a single scale of features from $10\%$ of input views (total only $^1/_{30}$ input of the full model), performance degradation is limited to $10\%$. This efficient approach even outperforms LERF [2] which utilizes all input views and scales, highlighting our method's robustness.

## 5 Limitations

The limitations of our method are twofold. First, unlike LERF [2], our method requires text labels before training. To perform segmentation with new text labels, our method needs to be retrained.

Inferring accurate boundaries with open vocabularies is challenging for implicit representations like NeRF, as NeRF learns a continuous representation rather than a discrete one. It is promising to learn object-level discrete representation using NeRF in future work.

Second, since our method has never seen any segmentation maps during training (only weakly supervised by the text labels), it fails to segment complex scenes like indoor datasets [70, 74] with high precision, as shown in the appendix. Our method distills pixel-level CLIP features in a patch-based fashion with a strong inductive bias for compact objects with an aspect ratio close to 1. For objects with large complex shapes and unobvious textures, our method would fail to recognize them.

## 6  Conclusion

In this study, we address the challenge of 3D open-vocabulary segmentation by distilling knowledge from the pre-trained foundation models CLIP and DINO into reconstructed NeRF in a weakly supervised manner. We distill the open-vocabulary multimodal knowledge from CLIP with a Selection Volume and a novel Relevancy-Distribution Alignment loss to mitigate the ambiguities of CLIP features. In addition, we introduce a novel Feature-Distribution Alignment loss to extract accurate object boundaries by leveraging the scene layout information within DINO features. Our method successfully recognizes long-tail classes and produces precise segmentation maps, even when supplied with limited input data, suggesting the possibility of learning 3D segmentation from 2D images and text-image pairs.

## Acknowledgement

We sincerely thank Zuhao Yang, Zeyu Wang, Weijing Tao, and Kunyang Li for collecting the dataset. This project is funded by the Ministry of Education Singapore, under the Tier-1 project scheme with project number RT18/22.

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

# Appendix

This document provides supplementary materials for *Weakly Supervised 3D Open-vocabulary Segmentation* in implementation details (Appendix A), more ablations (Appendix B), more evaluations (Appendix C), and more results (Appendix D).

## A    Implementation Details

### A.1    Model Architecture

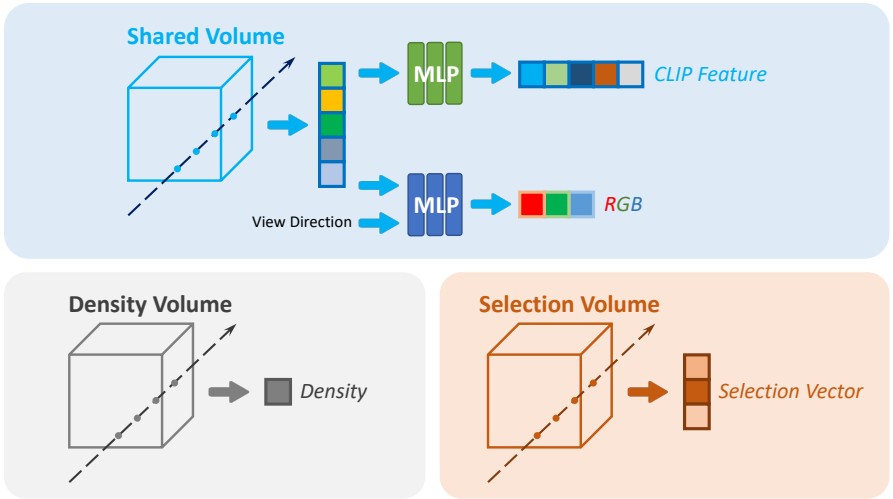

Figure 6: **Model architecture.**

We use TensoRF [42] as our base NeRF architecture for efficiency, the plane size is also the same as the default setting of TensoRF. The RGB and CLIP feature branches share the same volume and use the same intermediate features. The selection volume and density volume are two other independent volumes. We directly use the features extracted from the selection volume and density volume as the selection vector and the density value, as they have low dimensions and are view-independent. We use the original MLP architecture in TensoRF to extract the view-dependent RGB value and use another MLP which discards view direction input to extract the rendered CLIP feature. The architecture is illustrated in Fig. 6.

### A.2    Hyperparameters

We set $\tau = 0.2$ to get the shaper segmentation probability distribution $\acute{P}$. The offset $b$ is set to $0.7$ to measure the similarities of the DINO features, meaning that two DINO features are considered similar if their cosine similarity is larger than $0.7$, and different if less than $0.7$. We use 3 scales of CLIP features, and the patch sizes of each scale are set as $s/5, s/7$, and $s/10$, where $s$ is the smaller value in the width and height of the input image $I$. In the ablation studies, we use $s/7$ as the patch size of the single-scale CLIP feature input. The weights associated with similar and dissimilar DINO features in $\mathcal{L}_{FDA}$ are set as $\lambda_{pos} = 200$ and $\lambda_{neg} = 0.2$ by default. In certain scenes, we find that setting $\lambda_{neg}$ to 0.22 or 0.18 can produce better results. We use ViT-B/16 CLIP model to extract the image and text features and use version 1 dino_vitb8 model to extract the DINO features because it employs the smallest downsampling factor of 8 which is advantageous for high-precision segmentation.

### A.3    Training

To reconstruct a NeRF from multiview images of a scene, we follow the same training settings as TensoRF. For segmentation training, we train the model for 15k iterations. In the first 5k iterations,

Table 3: **Dataset.** We list the collected 10 scenes and the corresponding text labels. The background labels are in *Italic font*.

| Scene | Text Labels |
|-------|-------------|
| bed | red bag, black leather shoe, banana, hand, camera, *white sheet* |
| sofa | a stack of UNO cards, a red Nintendo Switch joy-con controller, Pikachu, Gundam, Xbox wireless controller, *grey sofa* |
| lawn | red apple, New York Yankees cap, stapler, black headphone, hand soap, *green lawn* |
| room | shrilling chicken, weaving basket, rabbit, dinosaur, baseball, *wood wall* |
| bench | Portuguese egg tart, orange cat, green grape, mini offroad car, dressing doll, *pebbled concrete wall*, *wood* |
| table | a wooden ukulele, a beige mug, a GPU card with fans, a black Nike shoe, a Hatsune Miku statue, *lime wall* |
| office desk | the book of The Unbearable Lightness of Being, a can of red bull drink, a white keyboard, a pack of pocket tissues, *desktop*, *blue partition* |
| blue sofa | a bottle of perfume, sunglasses, a squirrel pig doll, a JBL bluetooth speaker, an aircon controller, *blue-grey sofa* |
| snacks | Coke Cola, orange juice drink, calculator, pitaya, Glico Pocky chocolate, biscuits sticks box, *desktop* |
| covered desk | Winnie-the-Pooh, Dove body wash, gerbera, electric shaver, canned chili sauce, *desktop* |

we freeze the shared volume and density volume, and train the selection volume and the CLIP feature branch. For the rest 10k iterations, we further finetune the shared volume and the RGB branch. We use Adam optimizer with $betas = (0.9, 0.99)$. The learning rates for training the volume and MLP branch are respectively set to $0.02$ and $1e - 4$. For finetuning the volume and the MLP, the learning rates are set to $5e - 3$ and $5e - 5$. We also employ a learning rate decay with a factor of $0.1$.

The multi-scale pixel-level CLIP features of training views are pre-computed before training and the DINO features are computed with sampled patches on the fly during training. When computing $\mathcal{L}_{supervision}$ and $\mathcal{L}_{RDA}$, we randomly sample rays with a batch size of 4096. When computing $\mathcal{L}_{FDA}$ we randomly sample patches of size $256 \times 256$ with a batch size of 8. We use a downsampling factor of 8 when sampling rays and a factor of 5 when sampling patches. The model is trained on an NVIDIA A5000 GPU with 24G memory for $\sim$1h30min for each scene.

### A.4 Dataset

We capture 10 scenes using smartphones and use Colmap [75] to extract camera parameters for each image. We capture $20 \sim 30$ images for each scene and the resolution of each image is $4032 \times 3024$. We follow the data structure of LLFF [76]. We manually annotate the segmentation maps of 5 views for each scene as the ground truth for evaluation.

We list the text labels used in our experiments in Tab. 3. Note that we sometimes choose to add a general word to describe the whole background, such as *wall*, *desktop*, following LERF [2]. The text labels in our dataset contain many long-tail classes, which can be used to fully assess the open-vocabulary capability.

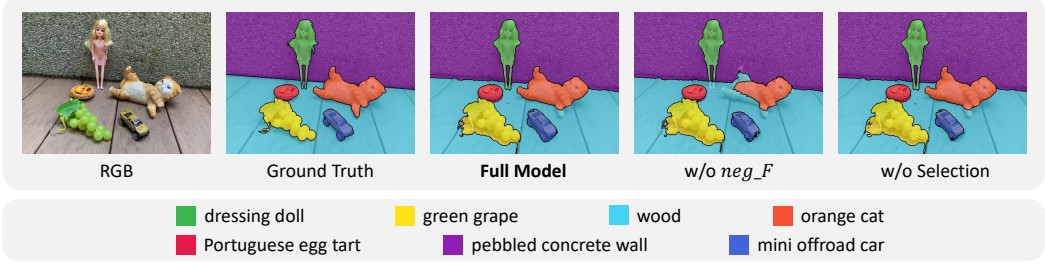

Figure 7: **More ablations.**

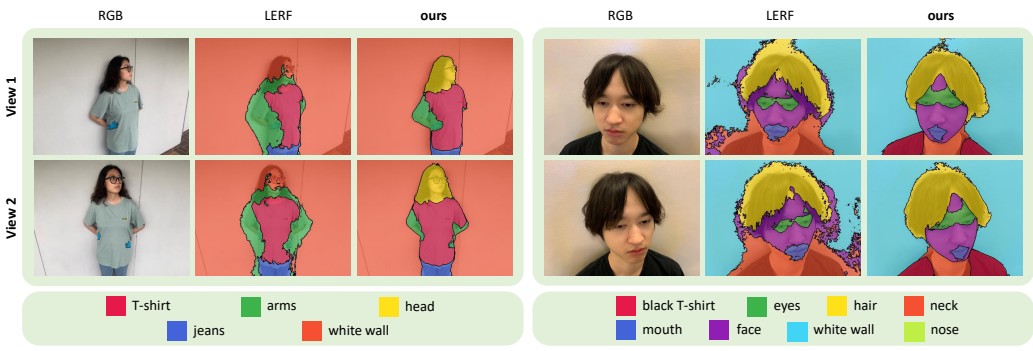

Figure 8: Evaluation on human body dataset (left). Evaluation on human head dataset (right).

## B  More Ablations

We perform two more ablation studies on the Selection Volume and the FDA loss, as shown in Tab. 4 and Fig. 7. Without the Selection Volume, we simply average the multi-scale CLIP features rather than learning the appropriate scale. We can see that both the mIoU score and the Accuracy score are inferior to the full model. We could discard the dissimilar part $neg\_F$ since dissimilar DINO features often impair the stability of the correlation loss. However, $neg\_F$ encourages different segmentation probabilities for different semantic regions and it plays a crucial role for precise object boundary extraction.

Table 4: **More Ablations.**

|  | mIoU | Accuracy |
| --- | --- | --- |
| w/o $neg\_F$ | 76.9 | 92.4 |
| w/o Selection | 84.8 | 95.3 |
| **full model** | **86.2** | **95.8** |

## C  More Evaluations

We additionally perform evaluations on human body, human head, indoor datasets with low-quality images [70, 74], and a complex scene from LERF datasets [2]. We compare with the concurrent work LERF qualitatively due to the lack of labels or the defective annotations as pointed out in [4]. We also perform experiments with different text prompts. We use the same scale level number and patch sizes in all comparisons.

**Human body and head.**  As shown in Fig. 8, our method segments more precise parts than LERF. Specifically, LERF fails to segment the "head" in the human body and the "black T-shirt" in the human head. In contrast, our method can recognize and segment these parts correctly because our designed RDA loss addresses the ambiguity of the CLIP features effectively.

**Indoor scenes with low-quality images.**  Fig. 9 shows experiments on the indoor datasets [70, 74], where many images are unrealistically rendered with less photorealistic appearances (as indicated in [4]) and have limited spatial resolution ($640 \times 480$ or $1024 \times 768$). Due to these data constraints,

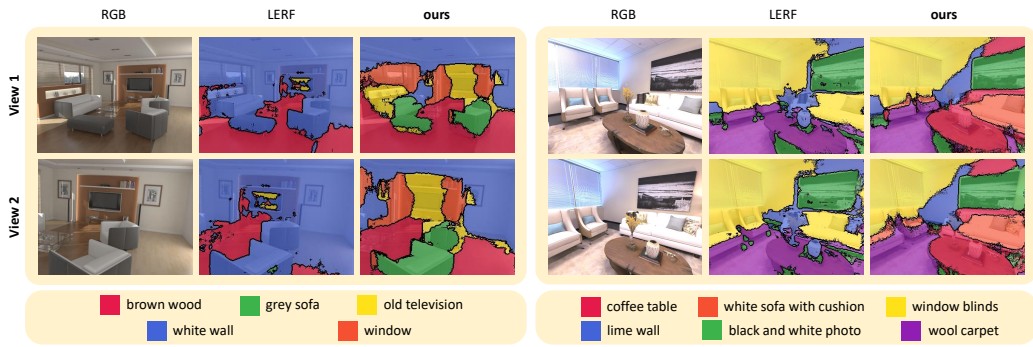

Figure 9: Evaluation on indoor datasets with lower quality images.

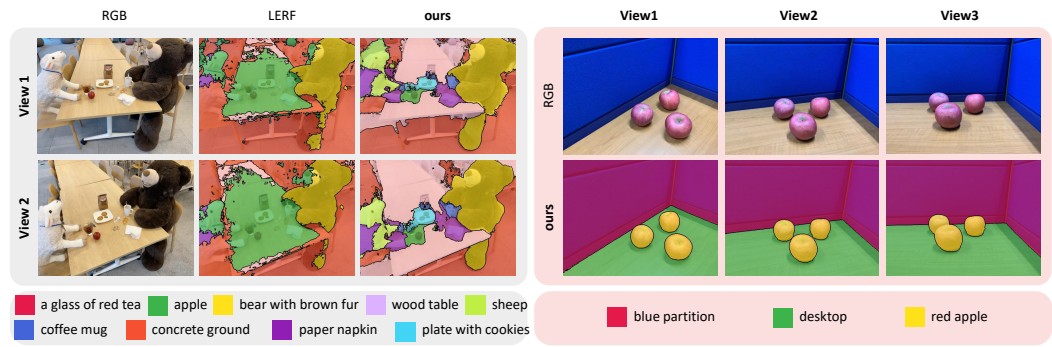

Figure 10: Evaluation on a complex scene from LERF datasets (left). Evaluation on a scene with multiple instances of a same class (right).

our method sometimes confuses labels with similar appearances. However, we can see that our method still outperforms LERF by successfully segmenting more labels.

**Complex scenes.** Fig. 10 (left) shows the segmentation of one challenging sample from the LERF dataset, where the scene has complex geometry as well as many objects of varying sizes. It can be observed that LERF cannot segment most objects due to the ambiguities of CLIP features while our method can segment more objects correctly with more precise boundaries. Fig. 10 (right) shows a scene with multiple instances of a same class. Since instances of the same class often share similar appearance, texture, etc., they also have similar DINO features. As a result, FDA will not mistakenly segment them into different classes. The RDA loss will further help by assigning all these instances to the same text label. In the experiment, we observed that our method successfully segments all three apples into the same class with accurate boundaries.

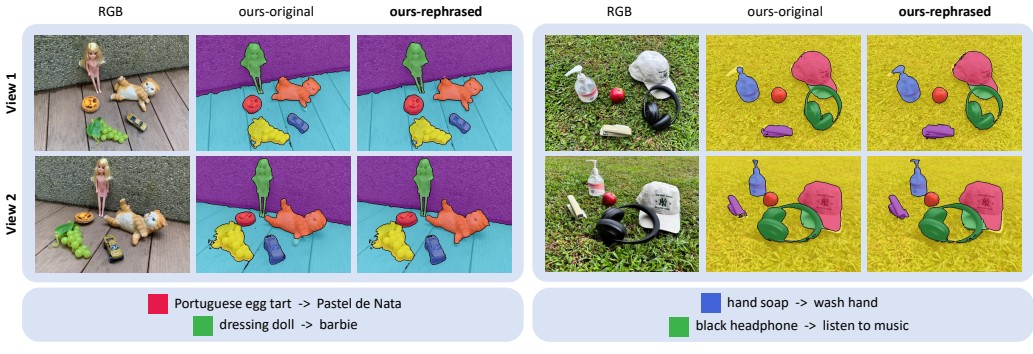

Figure 11: Evaluation on rephrased texts.

**Segmentation with different text prompts.**  We conduct experiments to segment scenes with different text prompts. In the experiments, we replaced the original texts with different languages (e.g., Portuguese egg tart -> Pastel de Nata), names (e.g., dressing doll -> Barbie), and actions (e.g., hand soap -> wash hand, black headphone -> listen to music). As Fig. 11 shows, with the rephrased text prompts, our method can still segment the scenes reliably. The experiments are well aligned with the quantitative experiments as shown in Tab. 5.

Table 5: Evaluation on rephrased texts.

|          | mIoU | Accuracy | mIoU | Accuracy |
|----------|------|----------|------|----------|
| original | 88.2 | 97.3     | 89.3 | 96.3     |
| rephrased| 89.3 | 97.2     | 88.4 | 96.6     |

# D   More Results

We show more segmentation visualizations of our method in Fig. 12 (bed), Fig. 13 (sofa), Fig. 14 (lawn), Fig. 15 (room), Fig. 16 (bench), Fig. 17 (table), Fig. 18 (office desk), Fig. 19 (blue sofa), Fig. 20 (snacks), and Fig. 21 (covered desk). The quantitative results are listed in Tab. 6.

Table 6: **Quantitative results.**

| bed | | sofa | | lawn | | room | | bench | |
|------|----------|------|----------|------|----------|------|----------|------|----------|
| mIoU | Accuracy | mIoU | Accuracy | mIoU | Accuracy | mIoU | Accuracy | mIoU | Accuracy |
| 89.5 | 96.7 | 74.0 | 91.6 | 88.2 | 97.3 | 92.8 | 98.9 | 89.3 | 96.3 |

| table | | office desk | | blue sofa | | snacks | | covered desk | |
|------|----------|------|----------|------|----------|------|----------|------|----------|
| mIoU | Accuracy | mIoU | Accuracy | mIoU | Accuracy | mIoU | Accuracy | mIoU | Accuracy |
| 88.8 | 96.5 | 91.7 | 96.2 | 82.8 | 97.7 | 95.8 | 99.1 | 88.6 | 97.2 |

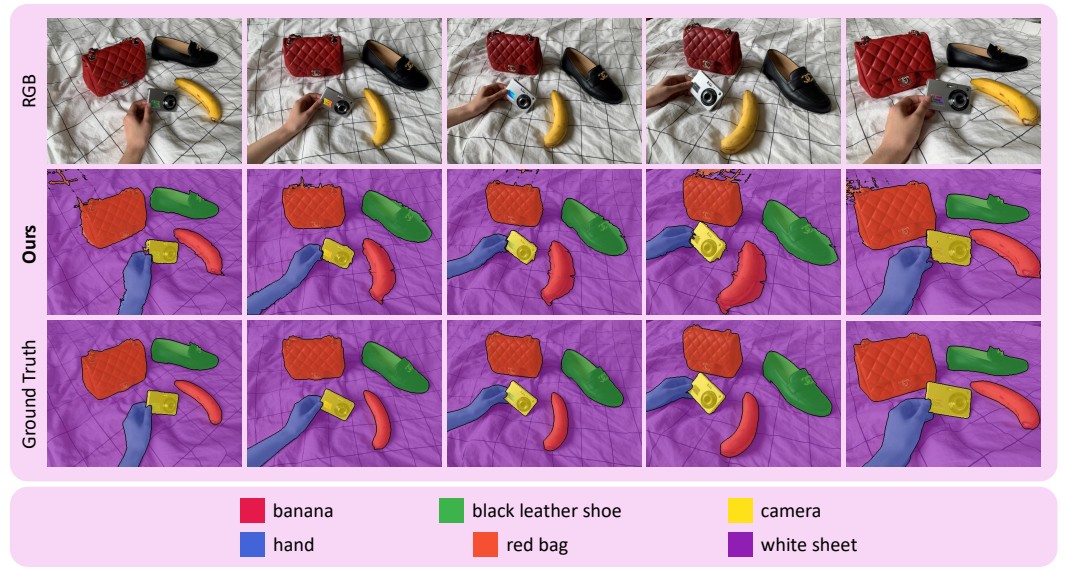

Figure 12: **bed.**

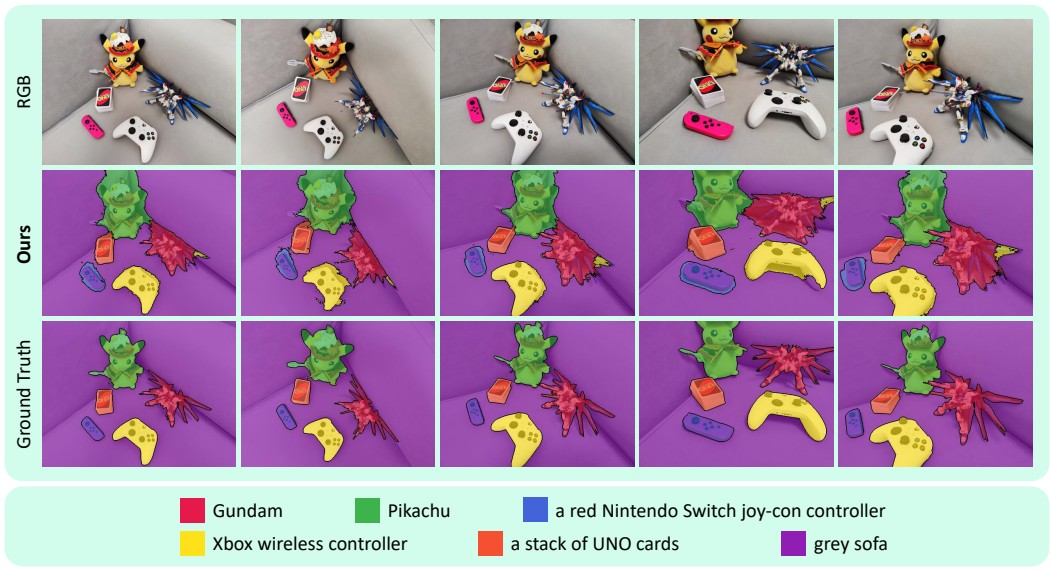

Figure 13: **sofa.**

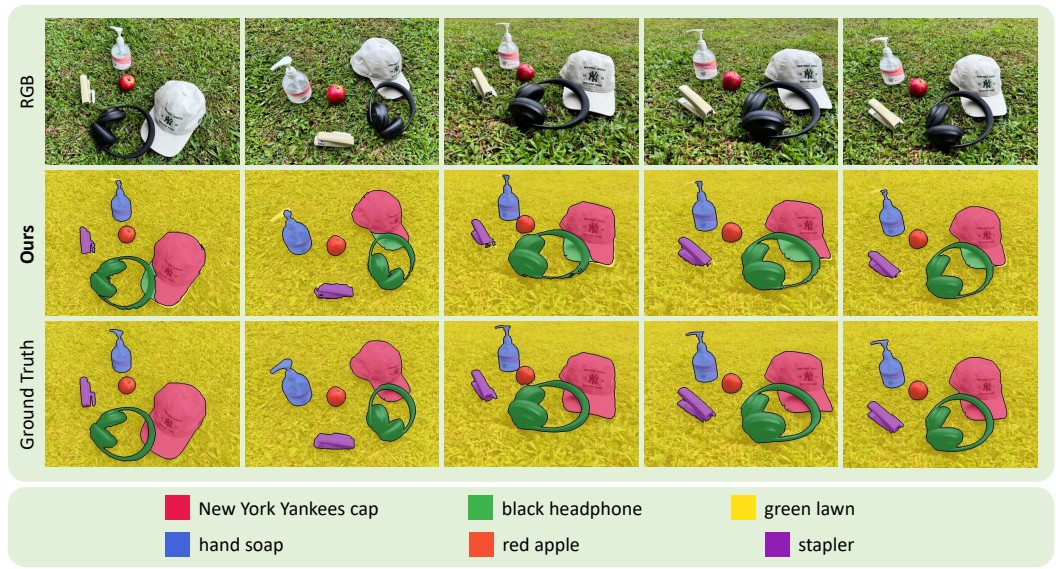

Figure 14: **lawn.**

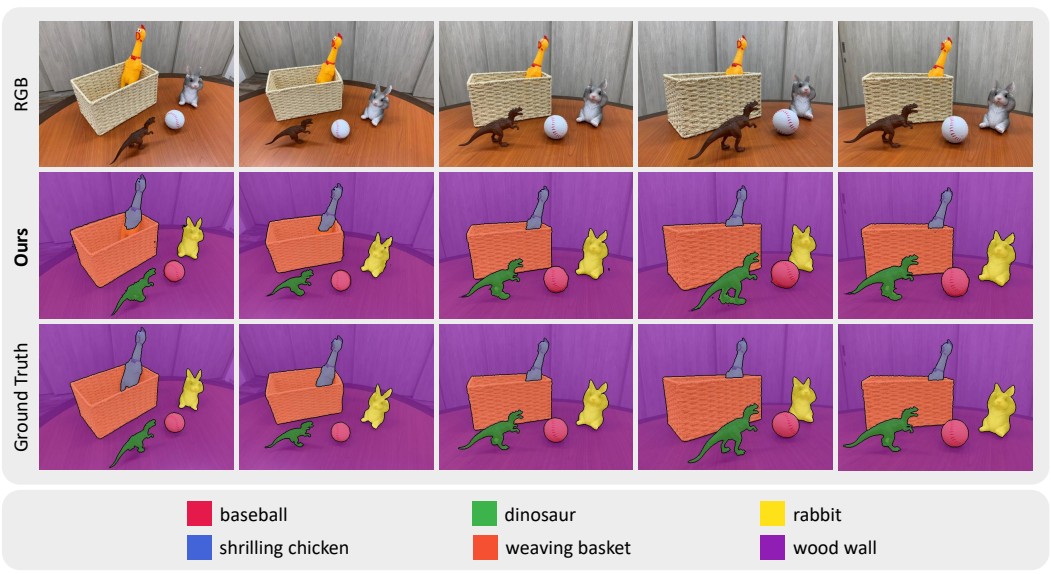

Figure 15: **room.**

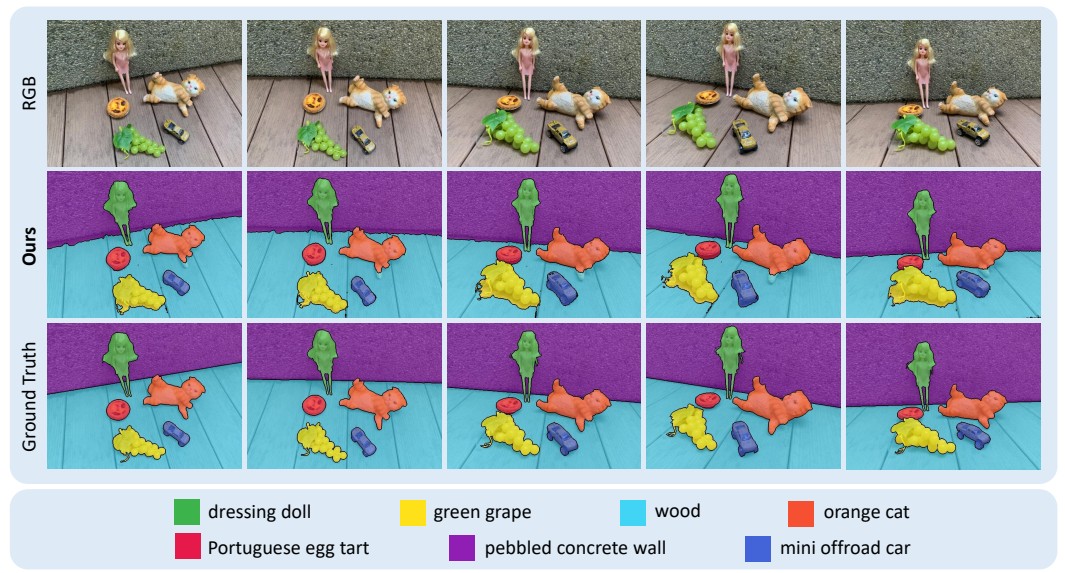

Figure 16: **bench.**

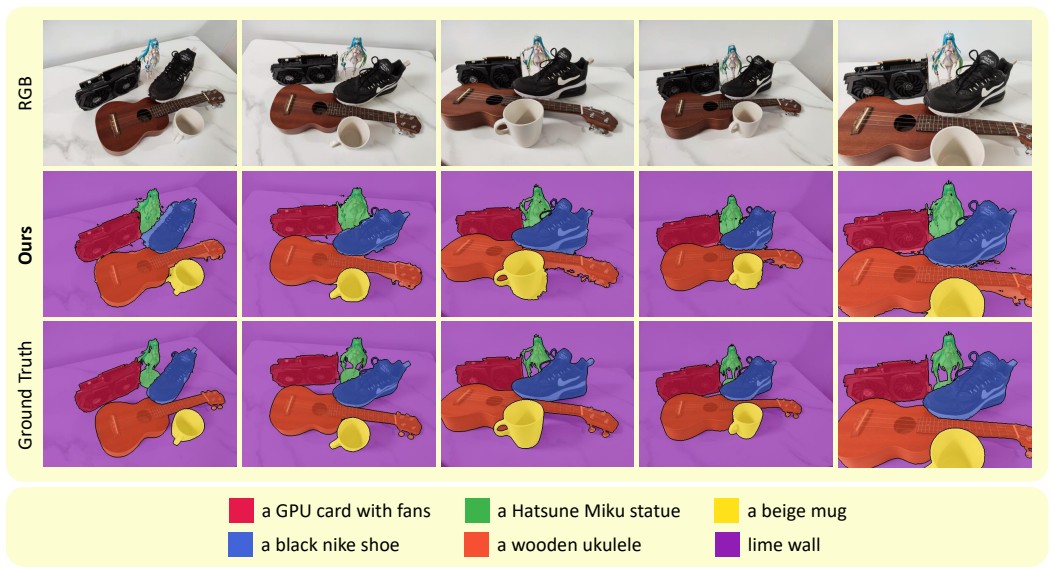

Figure 17: **table.**

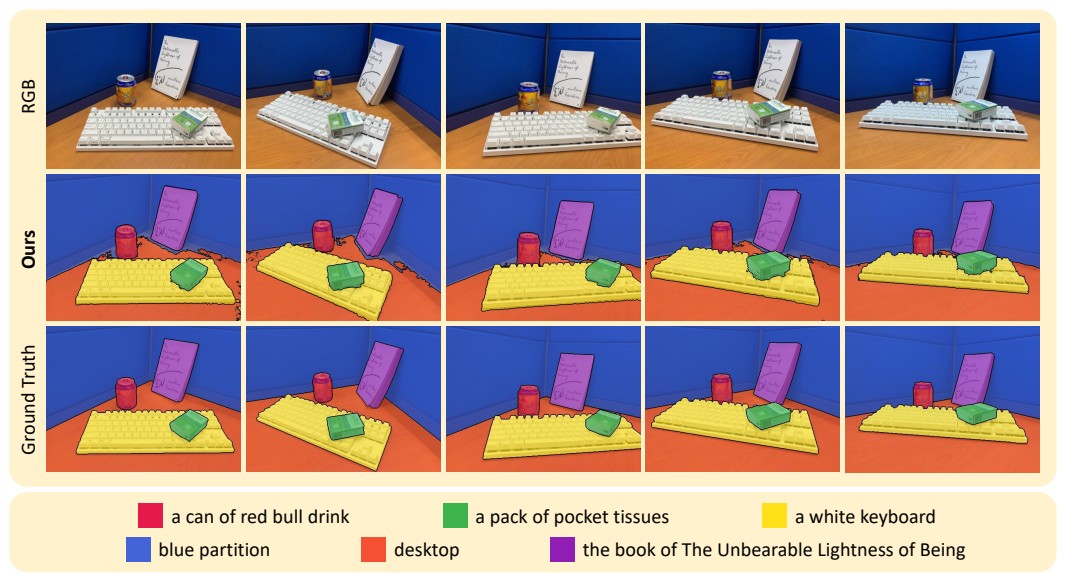

Figure 18: **office desk.**

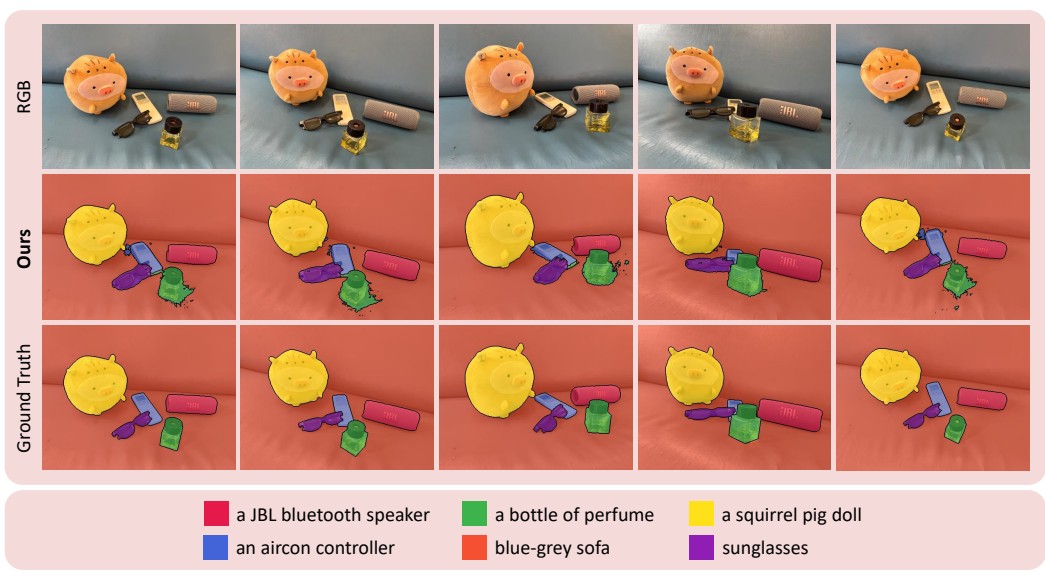

Figure 19: **blue sofa.**

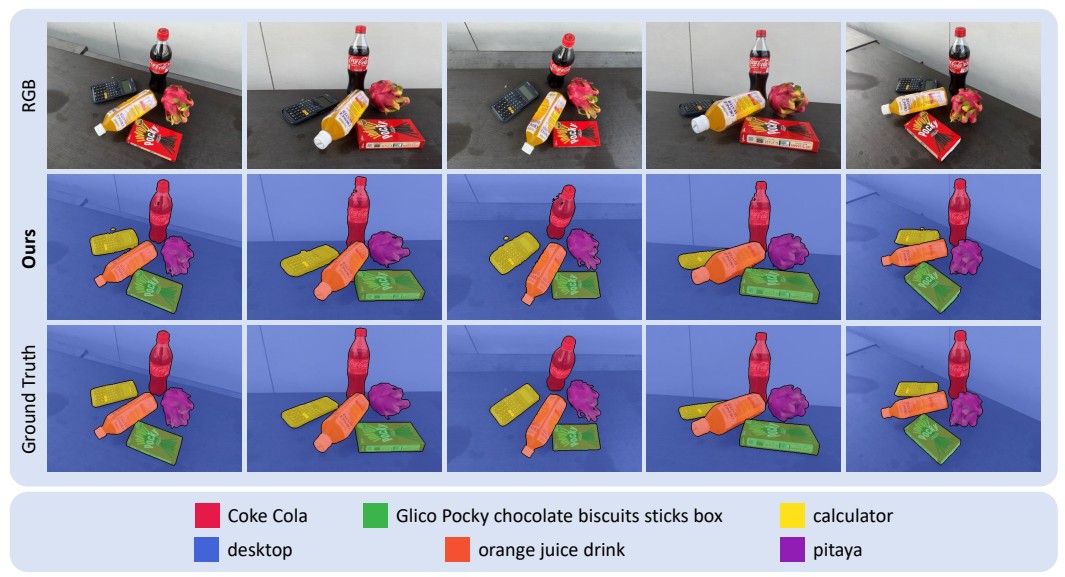

Figure 20: **snacks.**

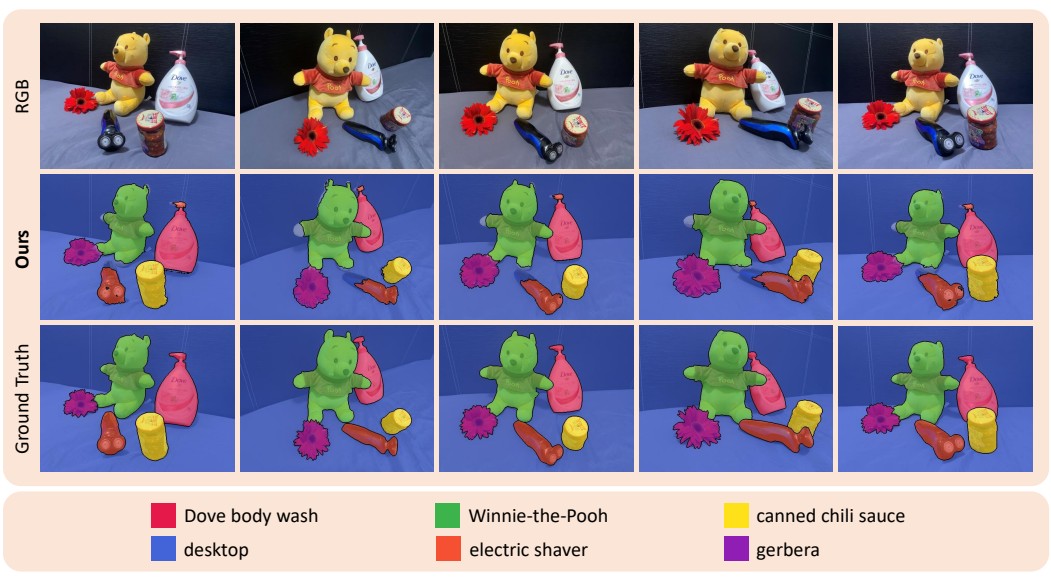

Figure 21: **covered desk.**

