| RGB | Ground Truth | **Full Model** | w/o $neg\_F$ | w/o Selection |

- 🟩 dressing doll
- 🟨 green grape
- 🟦 wood
- 🟧 orange cat
- 🟥 Portuguese egg tart
- 🟪 pebbled concrete wall
- 🟦 mini offroad car

Figure 2: **More ablations.**

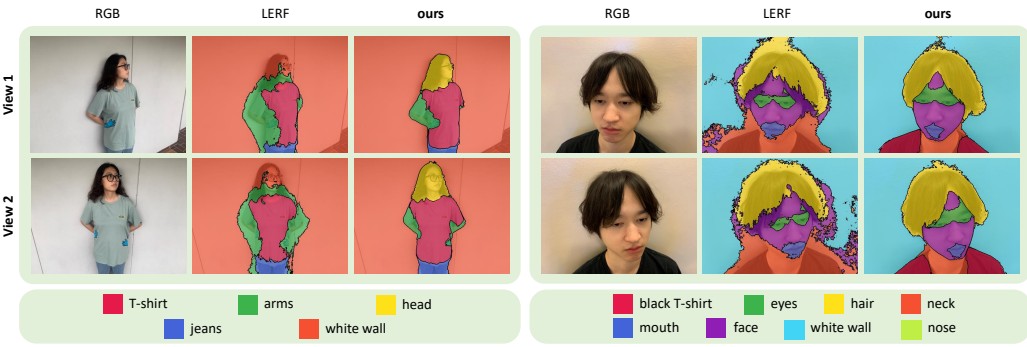

Figure 3: Evaluation on human body dataset (left). Evaluation on human head dataset (right).

## B    More Ablations

We perform two more ablation studies on the Selection Volume and the FDA loss, as shown in Tab. 2 and Fig. 2. Without the Selection Volume, we simply average the multi-scale CLIP features rather than learning the appropriate scale. We can see that both the mIoU score and the mAP score are inferior to the full model. We could discard the dissimilar part $neg\_F$ since dissimilar DINO features often impair the stability of the correlation loss. However, $neg\_F$ encourages different segmentation probabilities for different semantic regions and it plays a crucial role for precise object boundary extraction.

Table 2: **More Ablations.**

|  | mIoU | mAP |
| --- | --- | --- |
| w/o $neg\_F$ | 76.9 | 92.4 |
| w/o Selection | 84.8 | 95.3 |
| **full model** | **86.2** | **95.8** |

## C    More Evaluations

We additionally perform evaluations on human body, human head, indoor datasets with low-quality images [5, 6], and a complex scene from LERF datasets [4]. We compare with the concurrent work LERF qualitatively due to the lack of labels or the defective annotations as pointed out in [7]. We also perform experiments with different text prompts. We use the same scale level number and patch sizes in all comparisons.

**Human body and head.**    As shown in Fig. 3, our method segments more precise parts than LERF. Specifically, LERF fails to segment the "head" in the human body and the "black T-shirt" in the human head. In contrast, our method can recognize and segment these parts correctly because our designed RDA loss addresses the ambiguity of the CLIP features effectively.

**Indoor scenes with low-quality images.**    Fig. 4 shows experiments on the indoor datasets [5, 6], where many images are unrealistically rendered with less photorealistic appearances (as indicated in [7]) and have limited spatial resolution ($640 \times 480$ or $1024 \times 768$). Due to these data constraints,

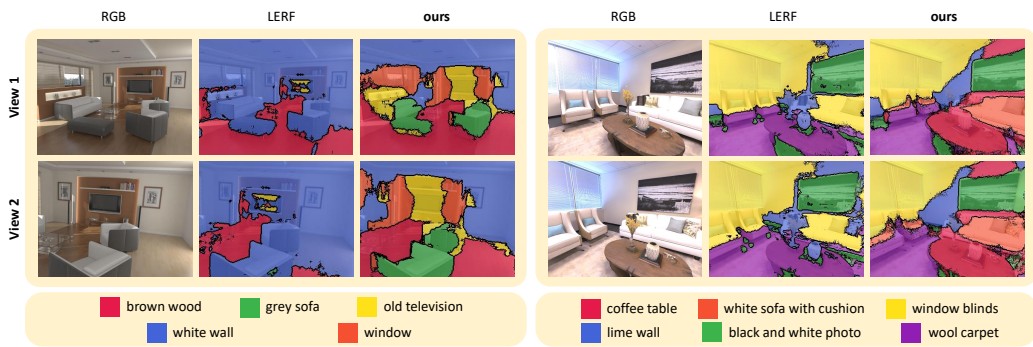

Figure 4: Evaluation on indoor datasets with lower quality images.

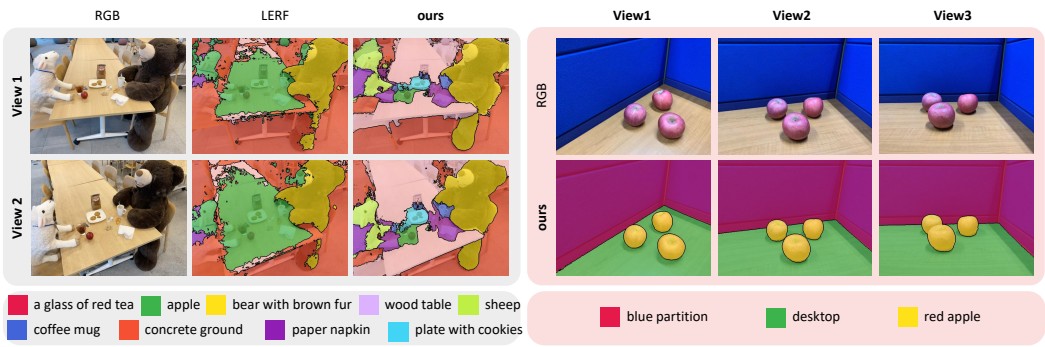

Figure 5: Evaluation on a complex scene from LERF datasets (left). Evaluation on a scene with multiple instances of a same class (right).

our method sometimes confuses labels with similar appearances. However, we can see that our method still outperforms LERF by successfully segmenting more labels.

**Complex scenes.** Fig. 5 (left) shows the segmentation of one challenging sample from the LERF dataset, where the scene has complex geometry as well as many objects of varying sizes. It can be observed that LERF cannot segment most objects due to the ambiguities of CLIP features while our method can segment more objects correctly with more precise boundaries. Fig. 5 (right) shows a scene with multiple instances of a same class. Since instances of the same class often share similar appearance, texture, etc., they also have similar DINO features. As a result, FDA will not mistakenly segment them into different classes. The RDA loss will further help by assigning all these instances to the same text label. In the experiment, we observed that our method successfully segments all three apples into the same class with accurate boundaries.

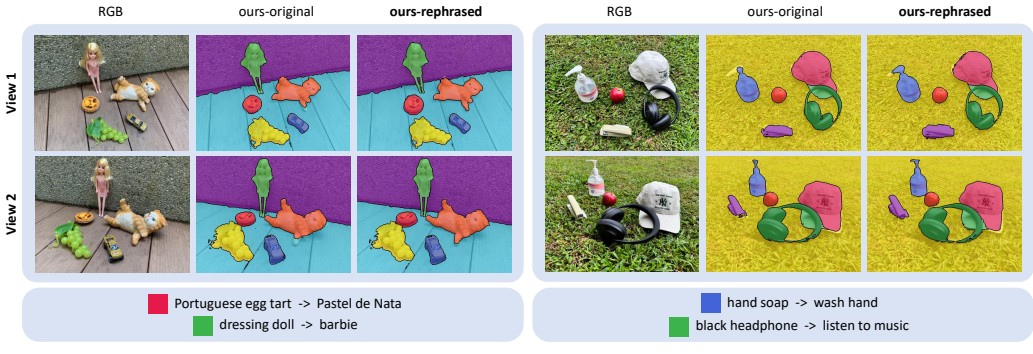

Figure 6: Evaluation on rephrased texts.

**Segmentation with different text prompts.** We conduct experiments to segment scenes with different text prompts. In the experiments, we replaced the original texts with different languages (e.g., Portuguese egg tart -> Pastel de Nata), names (e.g., dressing doll -> Barbie), and actions (e.g., hand soap -> wash hand, black headphone -> listen to music). As Fig. 6 shows, with the rephrased text prompts, our method can still segment the scenes reliably. The experiments are well aligned with the quantitative experiments as shown in Tab. 3.

Table 3: Evaluation on rephrased texts.

|  | mIoU | mAP | mIoU | mAP |
|---|---|---|---|---|
| original | 88.2 | 97.3 | 89.3 | 96.3 |
| rephrased | 89.3 | 97.2 | 88.4 | 96.6 |

## D   More Results

We show more segmentation visualizations of our method in Fig. 7 (bed), Fig. 8 (sofa), Fig. 9 (lawn), Fig. 10 (room), Fig. 11 (bench), Fig. 12 (table), Fig. 13 (office desk), Fig. 14 (blue sofa), Fig. 15 (snacks), and Fig. 16 (covered desk). The quantitative results are listed in Tab. 4.

Table 4: **Quantitative results.**

| bed | | sofa | | lawn | | room | | bench | |
|---|---|---|---|---|---|---|---|---|---|
| mIoU | mAP | mIoU | mAP | mIoU | mAP | mIoU | mAP | mIoU | mAP |
| 89.5 | 96.7 | 74.0 | 91.6 | 88.2 | 97.3 | 92.8 | 98.9 | 89.3 | 96.3 |
| table | | office desk | | blue sofa | | snacks | | covered desk | |
| mIoU | mAP | mIoU | mAP | mIoU | mAP | mIoU | mAP | mIoU | mAP |
| 88.8 | 96.5 | 91.7 | 96.2 | 82.8 | 97.7 | 95.8 | 99.1 | 88.6 | 97.2 |

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

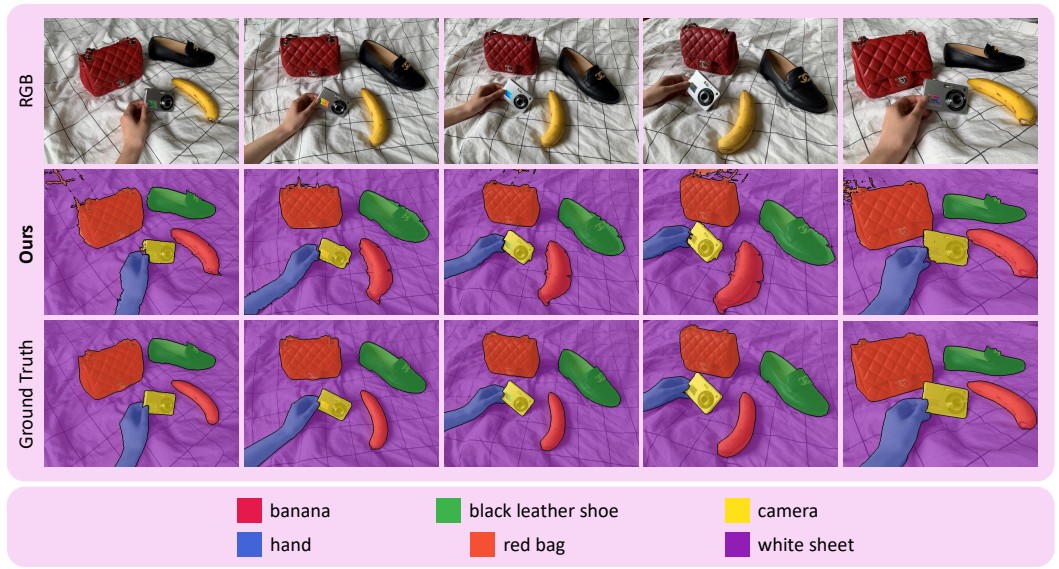

Figure 7: **bed.**

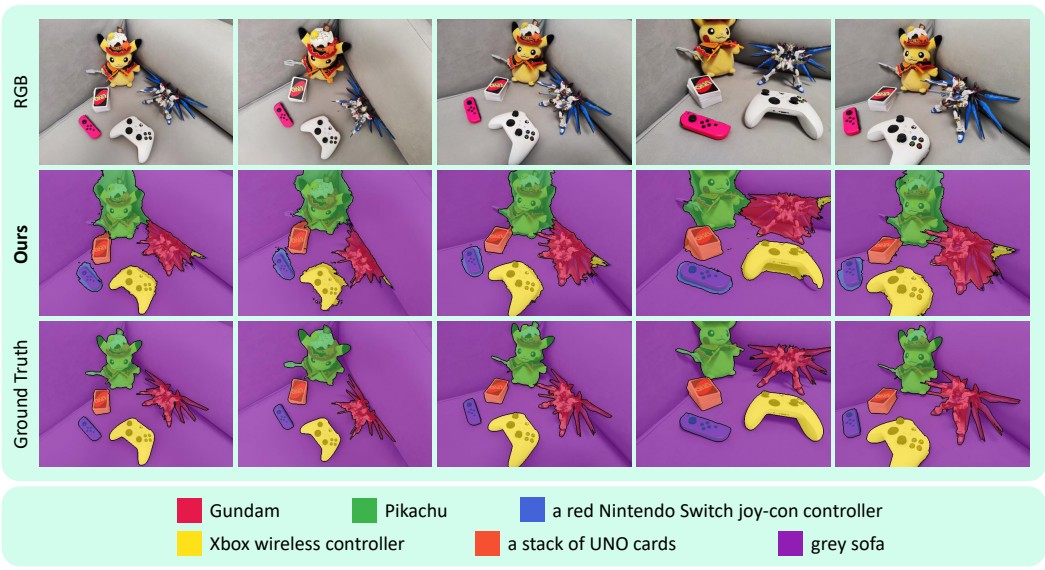

Figure 8: **sofa.**

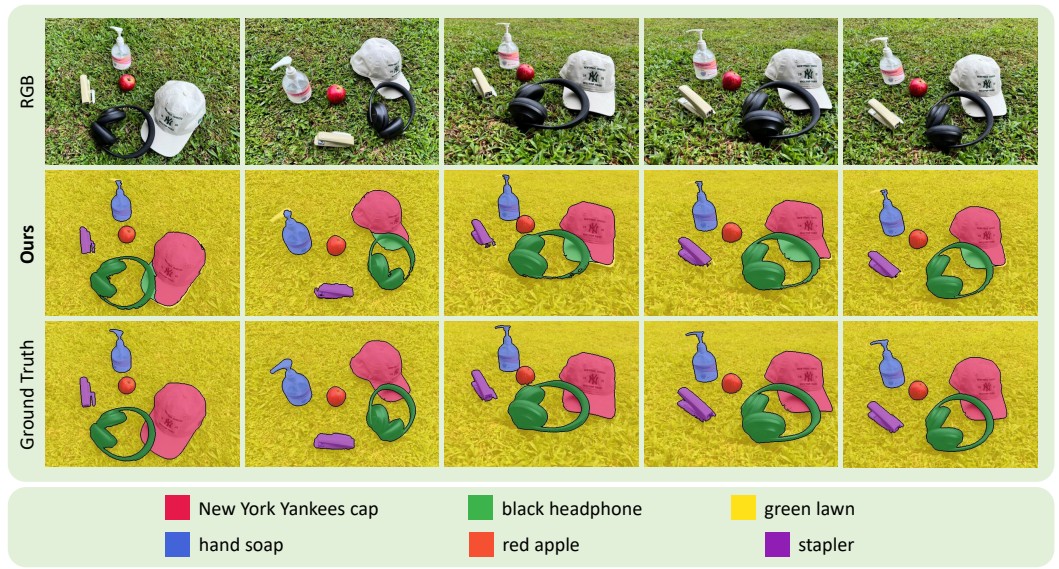

| | New York Yankees cap | | black headphone | | green lawn |
|---|---|---|---|---|---|
| | hand soap | | red apple | | stapler |

Figure 9: **lawn.**

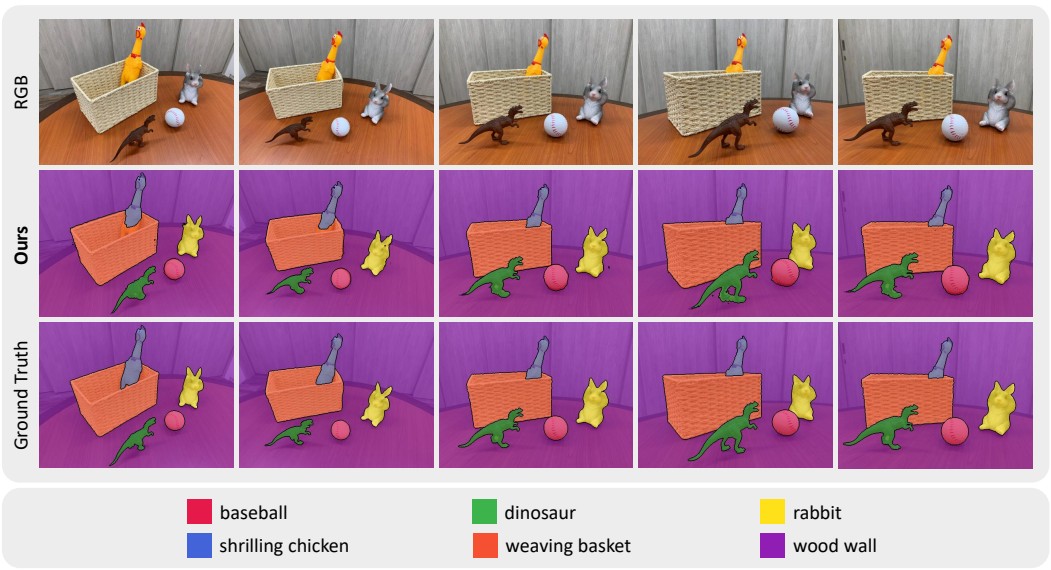

| | baseball | | dinosaur | | rabbit |
|---|---|---|---|---|---|
| | shrilling chicken | | weaving basket | | wood wall |

Figure 10: **room.**

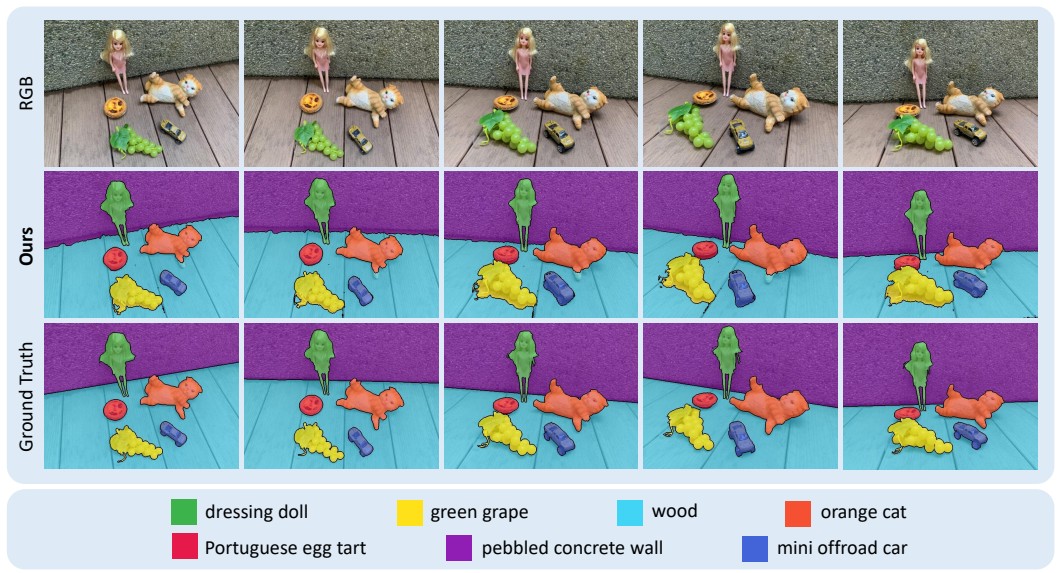

Figure 11: **bench.**

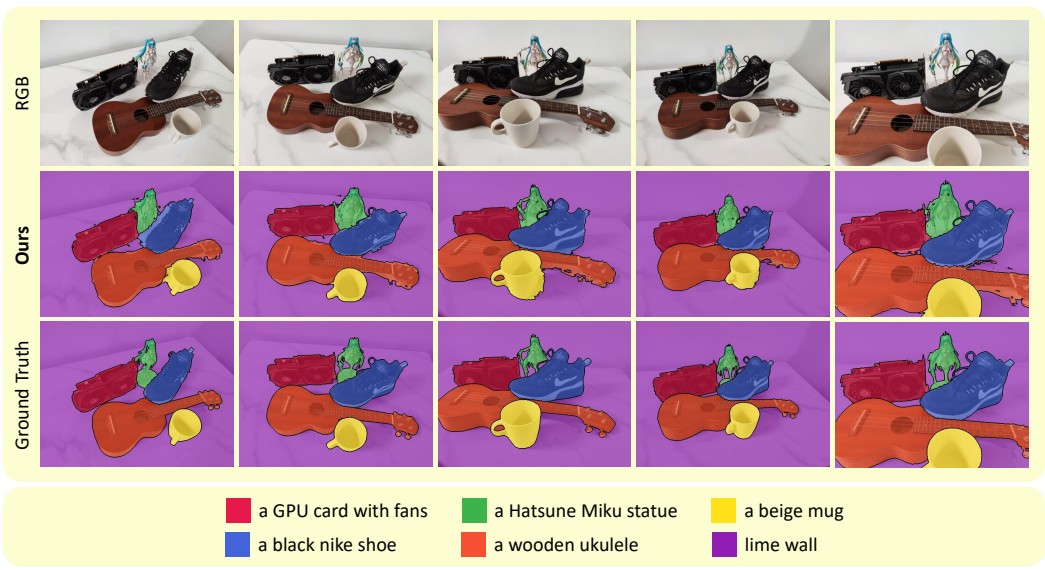

Figure 12: **table.**

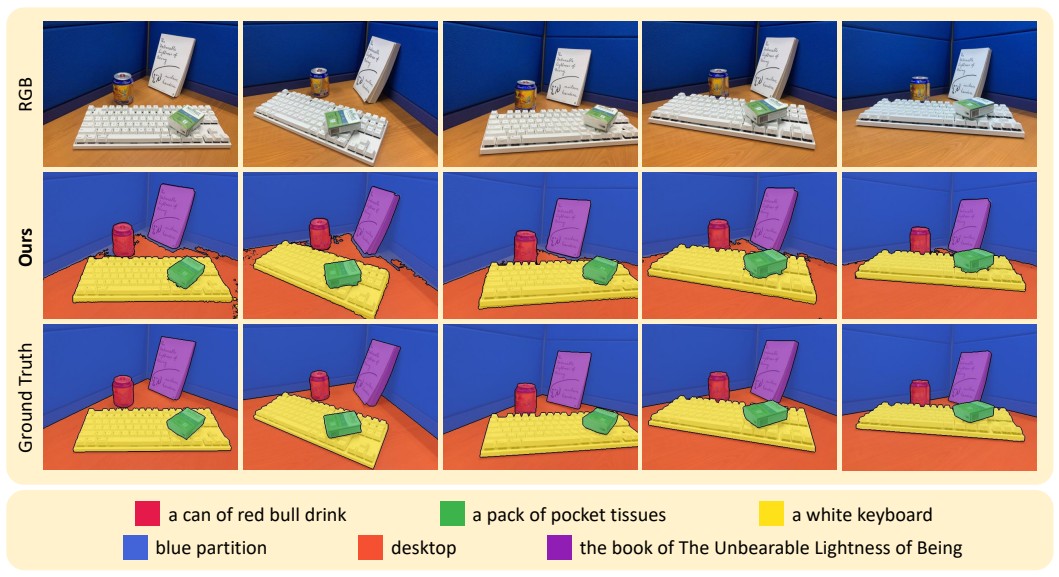

RGB

Ours

Ground Truth

a can of red bull drink
a pack of pocket tissues
a white keyboard
blue partition
desktop
the book of The Unbearable Lightness of Being

Figure 13: **office desk.**

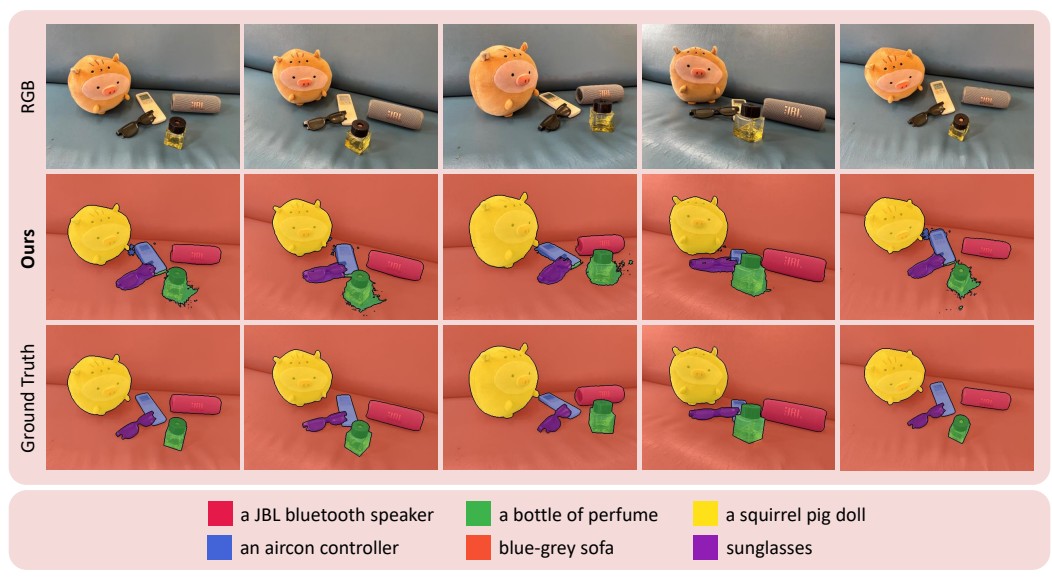

RGB

Ours

Ground Truth

a JBL bluetooth speaker
a bottle of perfume
a squirrel pig doll
an aircon controller
blue-grey sofa
sunglasses

Figure 14: **blue sofa.**

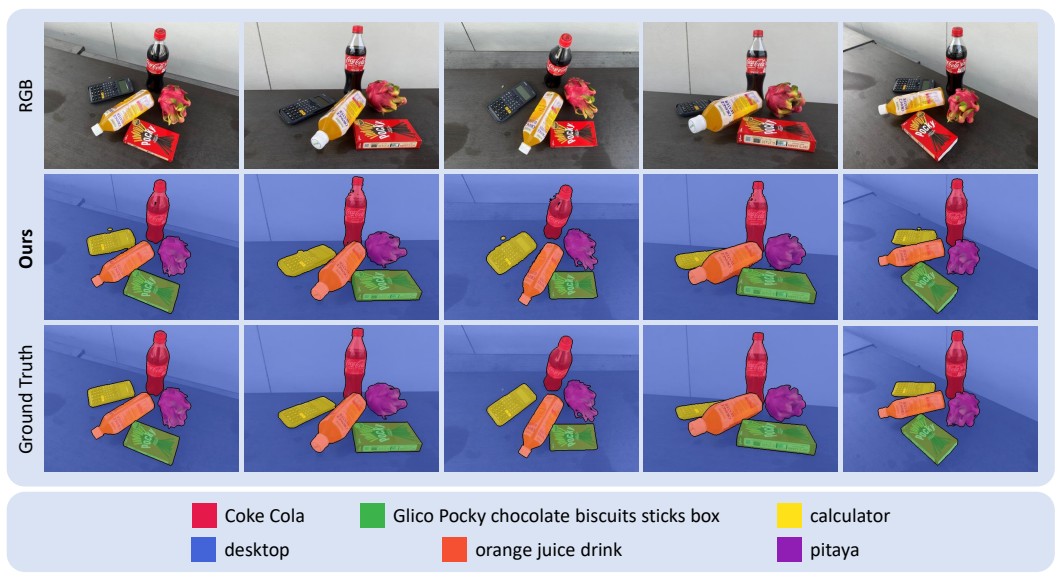

Figure 15: **snacks.**

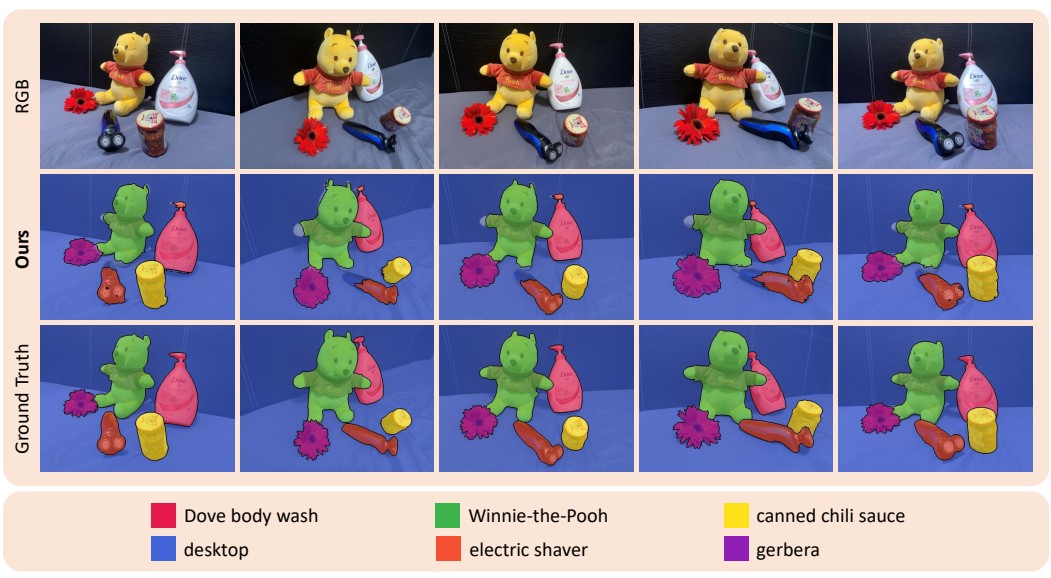

Figure 16: **covered desk.**