# OpenReview forum: "Weakly Supervised 3D Open-vocabulary Segmentation"
_NeurIPS.cc/2023/Conference — NeurIPS 2023 poster_

### Official Review · Reviewer_L8we · 2023-06-23

**Soundness:** 3 good
**Presentation:** 3 good
**Contribution:** 3 good
**Rating:** 6
**Confidence:** 4

**Summary:**

This paper proposes a new approach for fusing in 3D the output of 2D neural networks (CLIP an DINO) applied to the analysis of multiple views of a given static scene. The main application is open-vocabulary 3D segmentation of radiance fields. Compare to trivially fusing CLIP features, the method introduces a few improvements, such as re-normalising the response of CLIP and using DINO to regularise the segments. These improvements are shown to help segmentation significantly. Comparison to recent works, including concurrent work such as LERF, are included.

**Strengths:**

* Fusing open-ended features such as CLIP in 3D can be useful in applications, including, as demonstrated here, for open-vocabulary 3D segmentation.

* I appreciated the effort to compare to very recent *concurrent* work like LERF. The fact that the method outperforms the latter, at least in the authors' experiments, is a further bonus.

* There are a few ideas (class re-balancing, multi-scale feature selection, relevance renormalisation and consistency with DINO features) which are shown to improve significantly the performance compared to simply fusing CLIP features. This illustrates some difficulty with the latter approach, and also shows simple fixes to these issues.


**Weaknesses:**

* The idea of fusing 2D features in 3D reconstruction using neural fields is not particularly novel as such. Semantic NeRF has done it for segmentation, N3F and FFD for unsupervised features like DINO, Panoptic NeRF and others for semantic segmentation, etc. FFD already included some early experiment with open-ended language-based features (and this paper compares to them).

* The math needs significant revision. Some issues amounts to typos and inaccuracies, but others really make it hard to understand how the algorithm works in detail. Some aspects of the algorithm are also slightly surprising and need some discussion. See below:

     * Eq. (3) is the main loss that allows to learn the radiance field model, including its extension to predict the multi-scale CLIP features F and the scale selection vectors S. I understand most of this *except* why and how this can learn feature selection. First, the equation does not make literal sense. The inner product $\langle \hat F(\mathbf{r}) \cdot S(\mathbf{r}) F(\mathbf{r} \rangle$ misses a comma, but I expect that the intended read is  $\langle \hat F(\mathbf{r}) \cdot S(\mathbf{r}), ~ F(\mathbf{r} \rangle$. Second, the $\cdot$ product is unclear. $F$ is a $N_s \times D$ matrix and $S$ is a $N_s$-dimensional vector. Is this an element-wise product with broadcast? Thirdly, $\hat F$ are the multi-scale CLIP features "fused" (i.e., rendered) by the radiance field and $F$ the ones predicted from the 2D image. Why comparing these two weighed by $S$ should result in $S$ learning to select the appropriate scale of CLIP features to use for that point?

    * Eq. (6): Is this meant to be $\langle T, F_I \rangle$? The notation $\langle T \cdot F_I\rangle$ does not make literal sense. Furthermore, $T$ is a $C\times D$ matrix and $F_I$ is a $N_s\times D\times H\times W$ matrix. The output is $C\times H \times W$, so we can sort of guess what the authors mean to do, but the notation does not show how the scale dimension $N_s$ is reduced (presumably using the scale selector $S$, but the notation does not reflect that).

    * The RDA loss of Eq. (8) is a bit problematic. First, from a modelling perspective, it seems to be based on spatially re-normalizing the response of the CLIP features to all possible classes $c$ an a class-by-class basis (see also Fig. 2). This is done to prevent certain classes from "swamping" others. However, what happens if a class is *not* present at all in a scene? Wouldn't this *force* the class to still be segmented, thus incorrectly? Second, there are issues with Eq. (8) as such. The quantity $\bar R(\mathbf{r})$ is a $C$-dimensional non-negative vector, but, unless this is done implicitly, it does *not* sum to one. I.e., its is not a *probability* vector. Can you really interrupted Eq. (8) as a sum of JS divergences?

    * Eq. (9)  what is the notation $\cos f_{hw} \cdot f_{ij}$? Do you mean $\langle \frac{f_{hw}}{\|f_{hw}\|},\frac{f_{ij}}{\|f_{ij}\|} \rangle$  ?

    * What is $z$ in Eq. (9)?

    * I understand that the authors are trying to implement the segmentation distillation idea of [49] -- Eq. (12) certainly bears resemblance to that work. However, I had a really hard time understanding the notation used in Section 3.3 overall.

* The evaluation is only carried out on 10 custom scenes with comparatively little clutter. Given that the authors have been willing to compare to LERF (which is concurrent work, so this is definitely not required) I wonder if it would make sense to run the assessment on *their* 13 custom scenes as well.




**Questions:**

* See the technical issues above. Can you clarify the notation, and, especially, the question about renormalisation in the RDA loss?

* In the scaling experiment of Table 2, bottom 4 rows, you use a very small number of views (according to the text, just 2-3 -- se line 279) are sufficient to obtain good segmentation performance. Do you mean that you are still using all *RGB* views for 3D reconstruction and just 2-3 views with CLIP features, or that you are using only 2-3 features in total, including for 3D reconstruction? The latter would be very surprising.


**Limitations:**

There is no substantive discussion of limitations (e.g., no failure cases demonstrated, no section or paragraph discussing limitations). Note that the NeurIPS guidelines encourage explicitly the discussion of limitations.

There is no discussion of societal impact, ethics, copyright, data protection etc. I don't believe that any of this is particularly relevant to this paper though. I do not see direct potential harm stemming from this research, and the datasets used appear custom-collected and do not contain personal data.

---

> ### Author Rebuttal · Authors · 2023-08-06
>
> Thank you for your careful reading and valuable feedback. Below please find our clarification regarding your comments.
>
> ## Reply to weaknesses
>
> 1. **Novelty.** Please see the **General Response 3**.
>
> 2. **Math.**
>
> - In l.162 we use $cos \langle · \rangle$ to denote  cosine similarity which could cause confusion. We will modify it to $cos\langle,\rangle$. Thus the Eq.(3) becomes:
> $
> L_{supervision} = \sum_{\textbf{r} \in R} \left(
> \left \|  \hat{C}(\textbf{r}) - C(\textbf{r})\right \|_2^2 -
> \cos\langle\hat{F}(\textbf{r}) , S(\textbf{r})F(\textbf{r})\rangle
> \right).
> $
> The loss compares the rendered features with the selected features predicted from the 2D image.
> The Selection Volume is learnt under the supervision of both RDA loss and FDA loss, as the gradients of the two losses are back-propagated to the Selection Volume as well. The loss combination
>
> $
> L_{supervision} + L_{RDA} + L_{FDA}
> $
> allows the Selection Volume to learn the appropriate scale.
>
> - As mentioned before, after modification, Eq. (6) should be:
> $
> R_I = S_I \cos \langle T, F_I\rangle \in \mathbb{R}^{C\times H \times W}.
> $
> In Eq.(6) we ignore the spatial dimension $H,W$ for simplicity, which means that during computation, $T$ is broadcast to $C \times D \times H \times W$. Then the cosine similarity $\cos \langle T, F_I\rangle$ is performed on the first two dimensions, making $\cos \langle T, F_I\rangle$ a $C \times N_s \times H \times W$ matrix. Then $S_I$ is reshaped as $N_s \times 1 \times H \times W$. Then the $R_I$ is the dot product between $\cos \langle T, F_I\rangle$ and $S_I$ in the first two dimensions. The dot product is $C \times 1 \times H \times W$ which is then reshaped to $C\times H \times W$. **Because there are lots of details which can be better explained with code, we thus use such a simple notion here** in Eq.(6) to give readers a basic idea about how our algorithm works. We'll include the details in the appendix and release the code later.
>
> - First, our method is trained with a known set of classes in the scene, we'd like to add it to the limitations. It's worth mentioning that the baseline Sem(OV-Seg)[1] and Sem(ODISE)[1] also share this same limitation. They require the text labels before the training stage to obtain the segmentation maps used for training the NeRF. Second, before computing the Eq.(8), we apply a softmax function to $\bar{R}(\textbf{r})$ to make it a probability vector. We skipped this detail due to the space limit and we will include it in the appendix later.
>
> - As mentioned before, after modification, Eq.(9) becomes
> $
> \cos \langle f_{hw}, f_{ij} \rangle
> $.
>
> - $z$ is segmentation logits defined in Eq.(4).
>
> 3. **More experiments.** Please refer to the **General Response 1 and 2**. We have conducted experiments on more diverse datasets including human scenes, indoor scenes, and complex scenes.
>
> ## Reply to questions
> 1. Please see the reply to the weaknesses.
>
> 2. We use all views for 3D reconstruction and just 2-3 views with CLIP features in the studies with 10% views.
>
> [1] In-Place Scene Labelling and Understanding with Implicit Scene Representation

---

> > ### Comment · Reviewer_L8we · 2023-08-10
> > **Response to rebuttal**
> >
> > I thank the authors for clarifying their maths in the answer. I think I can now understand the details in a more rigorous manner, but their comment "Because there are lots of details which can be better explained with code, we thus use such a simple notion here" in the rebuttal leaves me perplexed. I understand the desire to simplify the notation, but using a notation which is suggestive but factually incorrect does not seem to me a great stylistic choice. It is possible to use a simple notation while carefully explaining its meaning to ensure that it isa also correct. After all, it took them just a few lines in the rebuttal to clarify this.
> >
> > The new evaluations, including comparison with LERF, are interesting -- and also go beyond what is required given that LERF is concurrent work.
> >
> > Overall, my inclination is to accept the paper.

---

> > > ### Comment · Reviewer_kCD4 · 2023-08-15
> > >
> > > Regarding the comparisons with (concurrent) LERF, and since I think it is not very clear from the paper and also marginalized in the rebuttal, I want to highlight again that the proposed methods relies on known object-classes for training the NeRF.
> > > This is not the case for LERF (and other baselines).
> > >
> > > I see two potential problems:
> > >
> > > - the comparison with LERF (and others) seems unfair as LERF does not make use of known object classes.
> > > - the method in fact cannot be called open-vocabulary since it relies on knowing the closed-set of object classes to optimize the NeRF.
> > >
> > > I am curious about your opinion on these points.

---

> > > > ### Author Response · Authors · 2023-08-16
> > > >
> > > > Thanks for your reply. Here we address your concern in the following three aspects:
> > > >
> > > > 1. **Our method significantly differs from close-vocabulary segmentation methods.** Close-vocabulary segmentation methods can only segment labels from a pre-defined set of classes in a dataset. When handling new labels, they require well-annotated training data at image, region, or pixel-level depending on the new labels [1]. Our method requires only the new labels, and it does not require any annotations of the training data (similar to applying CLIP under the zero-shot setting). It’s worth noting that, unlike 2D visual recognition tasks, there is no clear definition of open vocabulary in 3D counterpart due to the lack of large-scale 3D foundation models. The typical approach distills knowledge from 2D foundation models towards 3D tasks, in which there is no clear norm to avoid using text labels during the distillation.
> > > >
> > > >
> > > > 2. **Comparison with LERF[2].** Under the context of 3D open-vocabulary segmentation, LERF and our method require the same inputs for generating segmentation: pre-trained CLIP, DINO, multi-view images, and user-provided text labels. The user-provided text labels are also used for the same purpose – distilling knowledge from the CLIP. They are just used at different stages – LERF uses them at the inference stage and hence it does not acquire sufficient semantic knowledge for segmenting object boundaries accurately. The comparison is thus fair though the implementations of the two methods are not completely the same.
> > > >
> > > >
> > > > 3. **Comparison with the baselines Sem(ODISE)[3,4] and Sem(OV-Seg)[3,5].** We would highlight that our setup is identical to that of the two baselines which utilize user-provided text labels in the same manner as our method. Both baselines necessitate the target text labels to acquire 2D segmentation maps before NeRF training, and they are both considered open-vocabulary segmentation methods.
> > > >
> > > >
> > > > [1] Panoptic Lifting for 3D Scene Understanding with Neural Fields
> > > >
> > > > [2] LERF: Language Embedded Radiance Fields
> > > >
> > > > [3] In-Place Scene Labelling and Understanding with Implicit Scene Representation.
> > > >
> > > > [4] Open-vocabulary semantic segmentation with mask-adapted clip
> > > >
> > > > [5] Open vocabulary panoptic segmentation with text-to-image diffusion models.

---

> > > > > ### Comment · Reviewer_kCD4 · 2023-08-17
> > > > >
> > > > > 1. I only partially agree with this point. It is correct that closed-vocabulary methods can only segment labels from a pre-defined set of classes, but the same is true for the proposed method: The proposed losses **L_RDA and L_FDA can only be computed if the (closed) set of object classes is known** prior to optimizing the NeRF, cf. "method requires only the new labels" i.e. it *does* require training labels. As soon as a method requires a closed set of labels during training, we can no longer talk about open-set, specifically because it is no longer possible to query arbitrary objects from the trained scene representation during inference, thus **it is not an open-set method**, the trained NeRF is tailored to a closed-set of classes (and needs re-training for a different set of classes). However, I see that the method does not require annotated per-point labels, so **the method can be called weakly-supervised, or unsupervised**.
> > > > >
> > > > > 2. The claim that LERF and the proposed method require the same inputs is not accurate and potentially missleading: During training of the NeRF (generating the 3D scene representation), **the proposed method additionally requires a closed set of object classes during training**, i.e., the method only works if the object classes in the scene are known. This is not the case for LERF.
> > > > > The claim that the class labels *``are just used at different stages"* is exaclty the point: *it is somewhat similar to using test labels during training*. In particular it gives an unfair advantage to the proposed method which solves a much easier task of segmenting a scene into a known set of classes. However, methods like LERF are more general, they compute an open-set 3D scene representation during training, that can be queried for arbitrary objects during inference. The proposed method however would require retraining every time the queries change.
> > > > >
> > > > > 3. Comparison with ODISE and OV-Seg. *"Our setup is identical to the two baselines"* - This is incorrect: In Sec. 3.6. of ODISE, it is stated that *"the test categories may be different from the training ones"*. This is not the case for this approach since each scene is optimized during training for the test labels of that particular scene (via the L_RDA, and L_FDA). There is also no experiment that would demonstrate that the method is able to segment classes not seen during training.
> > > > >
> > > > > In particular, an important experiment is missing that would potentially highlight this point: How would the proposed method perform on an unseen set of classes? In particular, for the "table" scene shown in Fig.4, what is the performance reported in Table 1 when during training of the NeRFs (for both this method and LERF) only the "baseball" and "dinosaur" classes are used, and during inference it is evaluated on all 6 classes? This experiment would convincingly demonstrate that the proposed method is indeed an open-set method.

---

> > > > > > ### Comment · Reviewer_L8we · 2023-08-17
> > > > > > **Continued discussion**
> > > > > >
> > > > > > Hi, reviewer kCD4 makes a good point.
> > > > > >
> > > > > > I understand that the authors claim that LERF and their method can be seen as taking the same inputs (images + CLIP + queries) and produce the same output (3D segments). Conceptually, the equivalence is valid in this narrow sense, but kCD4 is correct that the two methods are not directly comparable in other important senses. Specifically, LERF does allow to query any object without the need for running optimisation again, which *is* an advantage that this paper does not enjoy.
> > > > > >
> > > > > > I do not think the this is a huge problem for this paper. LERF is concurrent work, and the comparison is valid in the narrow sense given above. The authors should, however, make this fact *very clear* in the final version, explicitly narrowing their claim in the comparison, including clearly communicating the advantages of LERF. I.e., do not just say "we outperform LERF in this metric", but also say something on the line of "but note that our method requires solving an optimisation problem for every new set of queries, whereas LERF can respond to new queries instantly".

---

> > > > > > > ### Comment · Reviewer_kCD4 · 2023-08-17
> > > > > > >
> > > > > > > Thanks for your comment **L8we**. Only to clarify, my main concern is not the comparison with LERF (which is concurrent work).
> > > > > > > My main argument is the fact that the approach is advertised as open-set / open-vocabulary (which it is not, as it depends on labels during training).
> > > > > > >
> > > > > > > This may sound philosophical but has important consequences:
> > > > > > > It gives an unfair advantage to the proposed method over the baselines (since the classes are known) which makes me question the meaningfulness of the experiments.
> > > > > > >
> > > > > > > I understand that this may sound like a subtle difference (using the object classes during optimization vs. using them only later during querying/inference) however, as mentioned already, I believe it is somewhat similar to mixing training and test data in supervised methods. I don't think this paper would set a good example for potential follow up works in the emerging field of  open-set 3D scene understanding.
> > > > > > >
> > > > > > > Nevertheless, I see merits in the method. I can imagine closed-set scenarios where the object classes are known in advance but no annotated training data is available -- however, in that particular scenario, the baselines have to be adapted accordingly for a fair comparison as well as the name/title of the method should be changed since this scenario does not correspond to open-set segmentation.

---

> > > > > > > ### Author Response · Authors · 2023-08-18
> > > > > > >
> > > > > > > We thank Reviewer L8we for sharing your view on the comparison with concurrent LERF. The following response to Reviewer kCD4 is below his review section.

---

> > > ### Author Response · Authors · 2023-08-16
> > > **Thanks!**
> > >
> > > We thank the reviewer for recognition of our paper.
> > >
> > > To make the notion more clear, we will update Eq. (6) to
> > > $
> > > R_{I_{hw}} = S_{I_{hw}} \cos \langle F_{I_{hw}}\cdot T \rangle,
> > > $
> > > where $h,w$ denotes the index in the $H$ and $W$ channel.

---

### Official Review · Reviewer_1xbk · 2023-07-05

**Soundness:** 2 fair
**Presentation:** 3 good
**Contribution:** 2 fair
**Rating:** 6
**Confidence:** 3

**Summary:**

This paper introduces a method that incorporates pre-trained CLIP features into Nerf for 3D open-vocabulary segmentation. The authors also utilize pre-trained DINO features to regularize the segmentation process and achieve precise segmentation boundaries. Additionally, they propose a Relevancy-Distribution Alignment loss to address ambiguities in CLIP features.

**Strengths:**

1. The paper demonstrates a fluent and well-structured organization.
2. The experimental results exhibit favorable performance.

**Weaknesses:**

1. The testing data is relatively simple, consisting of views with limited perspectives, and does not include results on larger and more complete 3D scenes (e.g., the benchmark proposed by Lerf).
2. Regarding the Feature-Distribution Alignment issue: If there are multiple objects of the same class, such as x1, x2, x3, in an image, their DINO features' similarity may be lower than b, but the Feature-Distribution Alignment could make them assigned to different classes, which is unreasonable. Although λ_neg is small, the supplementary material demonstrates the criticality of neg_F, and removing it significantly deteriorates the results.


**Questions:**

1. How would the performance be affected if the Relevancy-Distribution Alignment was not used during training and instead the segmentation probabilities were spatially normalized during inference?
2. How does the proposed method perform on Lerf's benchmark?

**Limitations:**

The processing of multi-spatial features of CLIP incurs substantial computational and storage resource requirements.

---

> ### Author Rebuttal · Authors · 2023-08-06
>
> Thank you for your valuable feedback. Below please find our clarification regarding your comments.
>
> ## Reply to weaknesses
> 1. **Experiments on other datasets including LERF dataset.** Please see the **General Response 1 and 2**. We have conducted experiments on more diverse datasets including human scenes, indoor scenes, and a complex scene from LERF datasets.
>
> 2. **Scene with multiple instances of a same class.** Please refer to Figure 3 (right) in the rebuttal PDF. We conducted a new experiment to demonstrate that our method can still perform well under such conditions. Since instances of the same class often share similar appearance, texture, etc., they also have similar DINO features. As a result, FDA will not mistakenly segment them into different classes. The RDA loss will further help alleviate this issue by assigning all these instances to the same text label. In the experiment, we observed that our method successfully segments all three apples into the same class with accurate boundaries.
>
> ## Reply to questions
> 1. **Deferred RDA.** Deferring RDA to the inference stage degrades the performance severely as shown in Table 1 below. Without RDA during training, the trained model yields lots of false label predictions due to the ambiguity of CLIP features, which further degrades FDA in discerning different classes.
>
> Table 1. Results with deferred RDA.
> | scene | *bench* |  | *sofa* | | *room* | | *table* | | *blue sofa* | |
> |-|-|-|-|-|-|-|-|-|-|-|
> |    | **mIoU** | **mAP**   | **mIoU**  | **mAP**   |**mIoU**  | **mAP**   |**mIoU**  | **mAP**   |**mIoU**  | **mAP**   |
> |deferred RDA|60.7|91.6|66.0|91.3|63.3|94.9|56.6|83.8|14.7|17.6|
> |**ours**    |89.3|96.3|74.0|91.6|92.8|98.9|88.8|96.5|82.8|97.7|
>
> | scene | *covered desk* |  | *bed* | | *lawn* | | *office desk* | | *snacks* | |
> |-|-|-|-|-|-|-|-|-|-|-|
> |    | **mIoU** | **mAP**   | **mIoU**  | **mAP**   |**mIoU**  | **mAP**   |**mIoU**  | **mAP**   |**mIoU**  | **mAP**   |
> |deferred RDA|41.4|58.4|75.6|94.6|70.3|91.3|38.3|55.6|58.3|89.6|
> |**ours**    |88.6|97.2|89.5|96.7|88.2|97.3|91.7|96.2|95.8|99.1|
>
>
> 2. **LERF Dataset.** Please see the **General Response 2**.

---

> > ### Comment · Reviewer_1xbk · 2023-08-14
> >
> > The response clearly solves my concerns. Thus, I improve my final rating from 5 to 6.

---

> > > ### Author Response · Authors · 2023-08-16
> > > **Thanks!**
> > >
> > > We thank the reviewer for appreciating our paper!

---

### Official Review · Reviewer_kCD4 · 2023-07-05

**Soundness:** 1 poor
**Presentation:** 3 good
**Contribution:** 2 fair
**Rating:** 2
**Confidence:** 4

**Summary:**

This paper tackles the task of 3D scene segmentation using CLIP and DINO distilled into a NeRF representation. Since CLIP and DINO are pre-trained, the method works in an unsupervised fashion i.e., no additional per-point training labels are needed. This is achieved by applying CLIP to small crops in a sliding window fashion resulting in pixel-aligned features. This is repeated for multiple scales, resulting in multiple pixel-aligned CLIP features maps, one for each scale. The paper introduces two new losses (FDA and RDA) that rely on a closed-set of classes during training of the NeRF. The method is evaluated on their own dataset and annotations, and compares to multiple, recent methods (LSeg, ODISE, OVSeg, LERF, FFD).

**Strengths:**

A novel “volume-selection” scheme used to align per-image CLIP features to per-pixel CLIP features, in addition to two new losses: Relevancy Distribution Alignment loss and Feature Distribution Alignment loss that do not require a dedicated pixel-aligned CLIP method.

**Weaknesses:**

I struggle a bit with the intermixing of the term “open-vocabulary” and the fact that the method relies on a known / closed set of classes C. E.g., “open-vocabulary text description for each class” (l.119) somehow implies that the set of classes that we want to segment is known a priori, whereas open-vocabulary implies that there are no such restrictions.
Importantly, the two losses presented in Sec.3.2 and Sec.3.3 seem to rely on a predefined set of classes C - i.e. **each NeRF is actually optimized for a closed-set of classes, and if the classes change, the NeRF needs to be retrained**. The method can be seen as open-world in the sense that there is no inherent limitation to the classes, however an optimized NeRF is a closed-set representation. Is that correct? I think this aspect should be discussed in more detail in the limitations.

The writing could be improved, numerous aspects are still unclear and should be properly introduced and explained (Please see the minor points under Questions - I consider them minor since it can be improved in an updated version, however there are so many points that are unclear that the overall presentation of the method is limitted).

This paper follows an unfortunate trend that is observable across the field of LLM-based methods for open-set scene understanding: each paper presents a new method and at the same time introduces a new (relatively small) evaluation dataset on which they improve over prior/concurrent work. It is clear that such evaluations are not entirely convincing since the results could be cherry picked and favoring the newly proposed model. This paper is unfortunately no exception to that trend. In particular, the dataset selection is motivated by the fact that ScanNet [5] is too limited in terms of classes - however there was a recent extension to ScanNet200 that should mitigate this problem. Also Replica [70] does feature 50-80 semantic classes (depending on the scenes) which shows a long-tail distribution. Hence, there are existing datasets (maybe not perfect) but at least established ones and certainly not worse than very few newly recorded scenes. Further, since the proposed method compares with LERF, it could also compare on the LERF dataset.

The argumentation / motivation in l.28-31 is unfortunate: it argues that the major challenge of open-set segmentation is the lack of datasets, however that is not really the case since open-set datasets cannot exist as it would require an infinite amount of all potentially existing labels.

The contributions (1) (in line 71-73) - how does it differ from Decomposing Nerf [4] ? They also use CLIP and DINO in a very similar fashion. Instead of LSeg one could use the proposed multi-scale CLIP feature maps and then compare, this would be a reasonable baseline.

**Questions:**

How is the selection vector S_x obtained? Eq. (2) shows that we can render it from the optimized NeRF, that is clear. However, we also need it already during the optimization, see Eq. (3). How does that work, i.e., how can the value S(r) that is only obtained via rendering of the optimized NeRF at the same time be used to optimize the NeRF?
Finally, I do not see an ablation study that shows the benefits of the 3D Selection Volume. Does the 3D selection volume improve over simply averaging multi-scale features as described in l.151?

Minor:

- l.4 “lack of [...] open-vocabulary segmentation datasets” I would even argue that open-vocabulary datasets are inherently impossible as it would require exhaustive labeling of all possible queries.

- l.7 ”finetuned with close-vocabulary datasets” this is only the case for a subset of visual-language models such as LSeg, other approaches are not fine tuned on closed sets of data e.g. OpenSeg.

- l.35 same point as before: OpenSeg is not fine tuned on a closed set of classes but weakly supervised via image captions.

- l.46 - ‘ambiguous similarities with text description’ what does that mean?

- l.53-55 I don’t understand this sentence, can you break it down for me? What are CLIP feature ambiguities? What is a segmentation probability distribution? What are class relevancies? Until l.62 is similarly unclear - I think it’s mostly unclear because the used terminology is not properly introduced / defined. I can guess what is trying to be achieved but that is not my job, it should be clearly written.

- l.126 “ambiguous similarities” - what does that mean?

- l.127 what are “relevancy values”? Fig.2 is not that helpful since it also does not explain the terms. Again, what is a segmentation probability distribution (l.130)? I can sort of guess it (maybe I’m wrong, maybe I’m right?), why not clearly define and explain these terms since a large part of the main contribution depends on them?

- l.169 Still unclear meaning of “ambiguities of the CLIP features”

**Limitations:**

Limitations are not addressed. For example, one could discuss the impact of the closed-set of classes C required to optimize the NeRF since this has implications on the practical application (see under weaknesses).

---

> ### Author Rebuttal · Authors · 2023-08-06
>
> Thank you for your careful reading and valuable feedback. Below please find our clarification regarding your comments.
>
> ## Reply to weaknesses
> 1. **Limitation.** Thank you for  your insightful comment. Indeed our method is trained with a known set of classes, we'd like to add it to the limitations. It's worth mentioning that the baseline Sem(OV-Seg)[3] and Sem(ODISE)[3] also share the same limitation. They require the text labels to first obtain the segmentation maps and they apply the maps for NeRF training.
> Training with unknown classes and extracting precise object boundaries at inference time is an extremely challenging task for 3D open-vocabulary segmentation on NeRF, which we will explore in future work.
>
> 2. **Dataset.** Please refer to the **General Response 1 and 2**.
>
> 3. **Major challenge of open-set segmentation.**
> We would like to respectfully highlight that, in our perspective, the major challenge of **3D** open-vocabulary segmentation is the lack of datasets.  In 2D, we have a large-scale dataset with 400M image-text pairs [1] which is deemed as open-vocabulary. In 3D, we do not have datasets of similar scale and vocabulary. ScanNet200 largely focuses on indoor scenes with common object labels only but lacks a wide range of diverse texts.
>
> 4. **Difference with FFD [2].** Please refer to the **General Response 3**.
>
> 5. **Proposed baseline.** We adopted a more advanced baseline LERF which can be viewed as your suggested baseline plus additional DINO supervision. In LERF, experiments show that the additional DINO supervision improves the precision of object boundaries clearly. Our method outperforms LERF both quantitatively and qualitatively, hence it should safely outperform your suggested baseline. This is well verified in the new experiments as presented in Table 1 below.
>
> Table 1. Comparisons with proposed baseline.
> | scene | *bench* |  | *sofa* | | *room* | | *table* | |
> |-|-|-|-|-|-|-|-|-|
> |    | **mIoU** | **mAP**   | **mIoU**  | **mAP**   |**mIoU**  | **mAP**   |**mIoU**  | **mAP**   |
> |proposed baseline|50.1|77.6|26.4|42.1|45.6|79.2|32.8|42.2|
> |**ours**         |89.3|96.3|74.0|91.6|92.8|98.9|88.8|96.5|
>
>
>
> ## Reply to questions
>
> 1. **Selection vector.**
> We use a NeRF model that has already been optimized through RGB reconstruction. The Selection Volume and feature branch are trained from scratch during the segmentation training stage. Therefore, the selection vector is learned in the segmentation training stage, instead of being obtained from an RGB-optimized NeRF. For more detailed information about the model architecture and training process, please refer to Appendix A.1 and A.3.
>
> 2. **Ablations.** We do have the ablation of the 3D selection volume, please see Appendix B for this ablation.
>
> Minor:
>
> 1. **Datasets.** We agree with you that collecting 3D open-vocabulary segmentation datasets is a very challenging task as it would require exhaustive labeling of a huge amount of 3D data.
>
> 2. **OpenSeg.**  Thank you for the information which will be updated in the Related Work. Nevertheless, we still believe that OpenSeg has smaller vocabulary and knowledge than CLIP as it is trained on a much smaller dataset.
>
> 3. **Ambiguous similarities.** Take an image patch with an apple lying on a lawn as an example (lawn is like class A and apple like class B in Fig 2 in the main paper). The corresponding CLIP feature contains both apple and lawn information. If the apple is relatively small, the patch could be classified as lawn as lawn dominates the patch's CLIP features. Thus the class apple would be ignored. Our proposed RDA loss helps address such ambiguity effectively.
>
> 4. **Relevancy and segmentation probability distribution.** Segmentation probability means the probability of a pixel belonging to a class. Relecancies mean the similarities between class text features and the images' CLIP features. These terminologies are defined in math in Eq. (5) and Eq. (6) of the main manuscript. We acknowledge the feedback and assure that we will thoroughly refine and highlight the terminology definition in the revised manuscript
>
> [1] Laion-400m: Open dataset of clip-filtered 400
> million image-text pairs.
>
> [2] Decomposing NeRF for Editing via Feature Field Distillation
>
> [3] In-Place Scene Labelling and Understanding with Implicit Scene Representation

---

> > ### Comment · Reviewer_kCD4 · 2023-08-15
> > **Thanks for the rebuttal!**
> >
> > Thanks a lot for the additional explanations and experiments that clarified some of my concerns.
> >
> > Indeed, your answer confirmed my concern as described in weakness 1: I believe we cannot say that relying on a known set of classes is a minor limitation. This is a fundamental flaw in a method that is titled "3D Open-Vocabulary Segmentation." If the object classes need to be known in advance, can it really be called open-vocabulary? I believe, if the object-classes need to be known during training/optimization of the NeRF representation then it is the same setting as closed-world 3D segmentation where all object-classes are known during training. I think this point is currently not clearly described in the paper and a bit hidden - what do the other reviewers think about this aspect?
> >
> > In fact, this strong dependence on knowing the object-classes also gives an unfair advantage to the proposed method over e.g. LERF in the comparison in Table 1 (as well as all qualitative results in the supplementary). I think this comparison is unfair since LERF does not use the known object-classes. Do I see this correct?
> >
> > I think this aspect needs to be discussed in more detail since not all reviewers seem to be aware of the limitation. I am therefore reducing my rating until this is discussed.

---

> > > ### Author Response · Authors · 2023-08-16
> > >
> > > Thanks for your reply. Here we address your concern in the following three aspects:
> > >
> > > 1. **Our method significantly differs from close-vocabulary segmentation methods.** Close-vocabulary segmentation methods can only segment labels from a pre-defined set of classes in a dataset. When handling new labels, they require well-annotated training data at image, region, or pixel-level depending on the new labels [1]. Our method requires only the new labels, and it does not require any annotations of the training data (similar to applying CLIP under the zero-shot setting). It’s worth noting that, unlike 2D visual recognition tasks, there is no clear definition of open vocabulary in 3D counterpart due to the lack of large-scale 3D foundation models. The typical approach distills knowledge from 2D foundation models towards 3D tasks, in which there is no clear norm to avoid using text labels during the distillation.
> > >
> > >
> > > 2. **Comparison with LERF[2].** Under the context of 3D open-vocabulary segmentation, LERF and our method require the same inputs for generating segmentation: pre-trained CLIP, DINO, multi-view images, and user-provided text labels. The user-provided text labels are also used for the same purpose – distilling knowledge from the CLIP. They are just used at different stages – LERF uses them at the inference stage and hence it does not acquire sufficient semantic knowledge for segmenting object boundaries accurately. The comparison is thus fair though the implementations of the two methods are not completely the same.
> > >
> > >
> > > 3. **Comparison with the baselines Sem(ODISE)[3,4] and Sem(OV-Seg)[3,5].** We would highlight that our setup is identical to that of the two baselines which utilize user-provided text labels in the same manner as our method. Both baselines necessitate the target text labels to acquire 2D segmentation maps before NeRF training, and they are both considered open-vocabulary segmentation methods.
> > >
> > >
> > > [1] Panoptic Lifting for 3D Scene Understanding with Neural Fields
> > >
> > > [2] LERF: Language Embedded Radiance Fields
> > >
> > > [3] In-Place Scene Labelling and Understanding with Implicit Scene Representation.
> > >
> > > [4] Open-vocabulary semantic segmentation with mask-adapted clip
> > >
> > > [5] Open vocabulary panoptic segmentation with text-to-image diffusion models.

---

> > > > ### Comment · Reviewer_kCD4 · 2023-08-17
> > > >
> > > > 1. I only partially agree with this point. It is correct that closed-vocabulary methods can only segment labels from a pre-defined set of classes, but the same is true for the proposed method: The proposed losses **L_RDA and L_FDA can only be computed if the (closed) set of object classes is known** prior to optimizing the NeRF, cf. "method requires only the new labels" i.e. it *does* require training labels. As soon as a method requires a closed set of labels during training, we can no longer talk about open-set, specifically because it is no longer possible to query arbitrary objects from the trained scene representation during inference, thus **it is not an open-set method**, the trained NeRF is tailored to a closed-set of classes (and needs re-training for a different set of classes). However, I see that the method does not require annotated per-point labels, so **the method can be called weakly-supervised, or unsupervised**.
> > > >
> > > > 2. The claim that LERF and the proposed method require the same inputs is not accurate and potentially missleading: During training of the NeRF (generating the 3D scene representation), **the proposed method additionally requires a closed set of object classes during training**, i.e., the method only works if the object classes in the scene are known. This is not the case for LERF.
> > > > The claim that the class labels *``are just used at different stages"* is exaclty the point: *it is somewhat similar to using test labels during training*. In particular it gives an unfair advantage to the proposed method which solves a much easier task of segmenting a scene into a known set of classes. However, methods like LERF are more general, they compute an open-set 3D scene representation during training, that can be queried for arbitrary objects during inference. The proposed method however would require retraining every time the queries change.
> > > >
> > > > 3. Comparison with ODISE and OV-Seg. *"Our setup is identical to the two baselines"* - This is incorrect: In Sec. 3.6. of ODISE, it is stated that *"the test categories may be different from the training ones"*. This is not the case for this approach since each scene is optimized during training for the test labels of that particular scene (via the L_RDA, and L_FDA). There is also no experiment that would demonstrate that the method is able to segment classes not seen during training.
> > > >
> > > > In particular, an important experiment is missing that would potentially highlight this point: How would the proposed method perform on an unseen set of classes? In particular, for the "table" scene shown in Fig.4, what is the performance reported in Table 1 when during training of the NeRFs (for both this method and LERF) only the "baseball" and "dinosaur" classes are used, and during inference it is evaluated on all 6 classes? This experiment would convincingly demonstrate that the proposed method is indeed an open-set method.

---

> > > > > ### Author Response · Authors · 2023-08-18
> > > > >
> > > > > Thanks for your exhaustive reply and suggestions. Here we have two more clarifications:
> > > > >
> > > > > **1**.We would like to reiterate that the baselines Sem(OV-Seg)[1,2] and Sem(ODISE)[1,3] also necessitate the use of text labels prior to NeRF optimization. The two baselines utilize Semantic-NeRF[1] for distillation, where Semantic-NeRF requires "some partial or noisy semantic labels for the images, such as ground truth labels for a small fraction of the images, or noisy or coarse label maps for a higher number of images." (Paragraph 5 of Section 1 in [1]). This can also be observed in their released codes at the link: https://github.com/Harry-Zhi/semantic_nerf.  Similarly, we feed Semantic-NeRF with the labels predicted by ODISE[2] or OV-Seg[3] (i.e., segmentation maps of input views), which are obtained by querying ODISE or OV-Seg with user-provided texts. Afterwards, Semantic-NeRF can be trained with the output segmentation maps. Hence, our networks are trained in the similar manner as the two baselines.
> > > > >
> > > > > Regarding the statement "the test categories may be different from the training ones" from [3], it only applies to the semantic segmentation in the 2D domain (i.e., an early step in Sem(ODISE)). To distill the 2D segmentation to 3D NeRF, [3] also needs the text labels before NeRF optimization.
> > > > >
> > > > >
> > > > > **2**.Regarding the naming of our method, as clarified in the previous response, there are no 3D open-vocabulary foundation models (to the best of our knowledge) so similar open-vocabulary setup and approach with 2D foundation models like CLIP is not feasible in 3D space. To achieve faithful 3D open-vocabulary segmentation with CLIP, our approach necessitates the class labels of the scene without which it’s almost impossible to distill precisely 'targeted' knowledge for the **dense prediction task** of semantic segmentation  (thus LERF only achieves very coarse segmentation). Hence, our work is not perfectly aligned with the definition of open vocabulary as in 2D space, but more an early exploration (and so one possible definition) of open vocabulary in 3D space while 3D foundation models are not available at this stage and even in the near future.
> > > > >
> > > > > We do agree with you that 'weakly-supervised' also defines our method well from a different perspective. We will adjust the naming of our method by either changing 'open-vocabulary' to 'weakly-supervised' as suggested or keeping 'open-vocabulary' but clearly indicating its differences from 2D open vocabulary and why we still call it open vocabulary.
> > > > >
> > > > >
> > > > > At last, we thank the reviewers again for your comments and insightful suggestion.
> > > > >
> > > > >
> > > > > [1] In-Place Scene Labelling and Understanding with Implicit Scene Representation.
> > > > >
> > > > > [2] Open-vocabulary semantic segmentation with mask-adapted clip
> > > > >
> > > > > [3] Open vocabulary panoptic segmentation with text-to-image diffusion models.
> > > > >
> > > > > [4] Panoptic Lifting for 3D Scene Understanding with Neural Fields

---

> > > > > > ### Comment · Reviewer_1xbk · 2023-08-19
> > > > > >
> > > > > > The author states, "To distill the 2D segmentation to 3D NeRF, ODISE [3] also needs the text labels before NeRF optimization." In fact, ODISE ultimately predicts n masks and their corresponding mask embeddings. Could you provide more details on how to obtain a complete 2D ODISE feature for an image?

---

> > > > > > > ### Author Response · Authors · 2023-08-19
> > > > > > >
> > > > > > > Note that Semantic-NeRF requires the final **segmentation maps** instead of **image features** to distill the segmentation for NeRF. To obtain the segmentation maps from ODISE, we utilize the `demo/demo.py` script provided in their released code (https://github.com/NVlabs/ODISE). We employ the user-provided text labels and execute `demo.run_on_image()` to extract `predictions["sem_seg"]`, which serves as the output 2D segmentation map. The size of `predictions["sem_seg"]` is `[n_class, H, W]`, representing the complete 2D segmentation map for an image.

---

> > > > > > > > ### Comment · Reviewer_1xbk · 2023-08-19
> > > > > > > >
> > > > > > > > Thanks for your reply.

---

### Official Review · Reviewer_t9cD · 2023-07-06

**Soundness:** 2 fair
**Presentation:** 3 good
**Contribution:** 2 fair
**Rating:** 4
**Confidence:** 4

**Summary:**

This work addresses the open vocabulary segmentation by leveraging pretrained foundation model, DINO and CLIP. They firstly lifts 2D CLIP features into NeRF. Then, relevancy distribution alignment loss is used to mitigate the CLIP features's ambiguity. Feature distribution alignment loss is used to improve object boundaries from DINO feature.

**Strengths:**

3D open vocabulary segmentation is an interesting and impactful topic. It is very challenging to use a fully supervised way for training such a model since annotation costs large amount of human labor. This paper leverages 2D foundation models. The presentation is fairly good and the proposed relevancy distribution alignment and feature distribution alignment losses are well motivated.



**Weaknesses:**

There are many unclear parts that make me feel hesitated to suggest acceptance of this paper.

1. how they used CLIP is not well justified. Will the patch-wise method break the CLIP feature space? I assume many patches covers only background or small portion of objects.

2. Why DINO? CLIP and DINO are two most important models for this work. So, motivating the utilization of them is important. Especially for DINO, why it is good for object boundaries. Visualization would help or ablation study will help.

3. the most concerning part is the experiments. The method is validated in limited settings. The class set is small that is not enough to show "open vocabulary" capability. The dataset is mall.



**Questions:**

See weaknesses.

**Limitations:**

The unclear parts in the paper and the limited experiments jointly make the performance of this work not convincing enough.

---

> ### Author Rebuttal · Authors · 2023-08-06
>
> Thank you for your valuable feedback. Below please find our clarification regarding your comments.
>
> 1. **CLIP features.** As many patches cover only the background or a small portion of objects, we use multi-scale CLIP features with varying patch sizes to mitigate this problem. The Selection Volume can be learned to pick sensible patch sizes to yield optimal segmentation. It is important to note that the patch-wise feature extraction is used in the concurrent LERF [1] as well, which achieves good results in recognizing objects using open-vocabulary text.
>
>
> 2. **Why DINO.** DINO demonstrates excellent performance in unsupervised segmentation [4] due to its ability to encode powerful, well-localized semantic information at high spatial granularity. Previous methods [2, 3] have shown that DINO can segment objects, parts, etc., with accurate boundaries. Additionally, DINO is well-suited for 3D segmentation because it produces continuous features across different views, as opposed to discrete segmentation maps. In contrast, 2D segmentation models often produce inconsistent segmentation maps across views, leading to the presence of artifacts in the final 3D segmentation, as depicted in Figure 4 of the main paper. Please refer to [2] for comprehensive visualizations of the DINO features.
>
>
> 3. **More experiments.** Please refer to the **General Response 1 and 2**. We perform additional experiments on more datasets and settings.
>
> [1] LERF: Language Embedded Radiance Fields
>
> [2] Deep ViT Features as Dense Visual Descriptors
>
> [3] NeRF-SOS: Any-View Self-supervised Object Segmentation on Complex Scenes
>
> [4] Unsupervised Semantic Segmentation by Distilling Feature Correspondences

---

> > ### Comment · Reviewer_t9cD · 2023-08-19
> >
> > Lack of comprehensive quantitative evaluation is a big concern. Although the authors added more evaluations, they are all qualitative. It still cannot improve my level of confidence about the performance of the proposed method. So, I will keep my original rating.

---

> > > ### Author Response · Authors · 2023-08-20
> > >
> > > From the qualitative results, our method outperforms LERF by a large margin. It recognizes more objects and segments more precise boundaries, which demonstrates the superior performance of our method.
> > >
> > > We would also like to remind the reviewer that the LERF dataset lacks segmentation annotations, and the indoor datasets "suffer from defective annotations" and "some objects are difﬁcult to predict their labels due to label ambiguity." This issue necessitates the "fixing of the label set of the dataset" for proper evaluation (as mentioned in Appendix C of FFD[1]). Consequently, quantitative evaluations on these datasets require exhaustive reannotations. We will add the quantitative results once we have completed the reannotations.
> > >
> > > [1] Decomposing NeRF for Editing via Feature Field Distillation

---

### Official Review · Reviewer_1C3c · 2023-07-09

**Soundness:** 4 excellent
**Presentation:** 3 good
**Contribution:** 3 good
**Rating:** 5
**Confidence:** 4

**Summary:**

This paper trained a NeRF model leveraging on two self-supervised trained models CLIP and DINO. By distillating multi-view image features from these two large models, the NeRF model is able to synthesize accurate object segmentation masks on unseen scenes with long-tailed distributions. Experiments on 10 self-collected scenes shows that high-quality segmentation masks can be predicted after per-scene training without explicit segmentation annotations.

**Strengths:**

- This paper proposes a easy-to-follow pipeline to reach accurate open-world 3D segmentation by taking advantages of two pretrained large models CLIP and DINO without access to segmentation ground truth masks.

- The experiments on self-collected scenes prove the capability of proposed method, and the ablations cover most of key design choices made in the paper, with adequate implementation details.

**Weaknesses:**

- Limited experiments on popular benchmarks.

    My main concern lies on the limited evaluation as 'open world 3d segmentation' is a challenging topic.
    - In this paper, 10 scenes with multi-view high-res images are taken for experiments. However, more diverse results on different datasets are expected to really prove the capability of ‘open-world 3D segmentation’. Existing datasets like ScanNet, VG150 still contains imbalanced long-tail distributions. How the proposed methods work on ScanNet, for example, where captured images are in VGA resolution and lower quality compared to used data in the paper.
   - There are many hyper-parameters need to be considered during training. Are they sensitive to difference scenes/datasets?

    Otherwise, the overall experiments may look too curated to prove the main statements.


 - One question to ask is whether the model sacrifice the ‘close-world’ capability to complement the open-world one. Therefore, it would be good to see how trained methods work on different categories with different types of text prompts.


- What is the impact of text descriptions? Are the given text carefully chosen or some general text input covering key information are good enough for generating promising labels?


Other comments:
- Why only 6 of 10 scenes are shown for numerical experiments with baseline methods and 4 scenes are used for ablations? I see a complete 10 scene results of proposed methods in the supplement but not of baselines.
- One question rather than weakness. Could authors explain why to choose to learn CLIP features with a shared TensorRF volume with colour instead of separately. Is there any specific reason or ablations for such design choice?

**Questions:**

Please see weaknesses above where my main concern is the lack of analysis on diverse popular real-world scenes to really prove the model's capability.

Overall this paper is attempting to tackle a challenging task and I think more discussion could better  strengthen the contributions.

**Limitations:**

Limitations and potential negative societal impacts have been discussed in the supplement.

---

> ### Author Rebuttal · Authors · 2023-08-06
>
> Thank you for your valuable feedback and pointing out we are attempting to tackle a challenging task. Below please find our clarification regarding your comments.
>
> 1. **Limited experiments.** Please refer to the **General Response 1 and 2**. We perform additional experiments on indoor datasets as well as other datasets.
>
> 2. **Hyperparameters** We would like to clarify that our method is not sensitive to the hyperparameters in most evaluated scenes. Specifically, we use the same hyperparameters in all experiments, except the *office desk* scene where we set $\lambda_{neg}$ to 0.35 which we found gives the best results. Using the default parameter setting, the score of this scene is mIoU=76.3, mAP=90.1, which still outperforms LERF as shown in Table 1 below.
>
> 3. **Text prompts.** Please refer to the **General Response 2**. We provide quantitative and qualitative results with more diverse text labels.
>
> 4. **More results.** We conducted experiments on part of the datasets due to limited computation resources. As suggested, we conducted experiments on the remaining scenes and reported the quantitative results of baselines of all scenes in Table 1. We also reported the core ablations about RDA and FDA in all scenes in Table 2. We will include these new experiments in the updated manuscript. Thank you for your suggestion.
>
> Table 1. Results of baselines on the remaining scenes.
> | scene | *blue sofa* || *covered desk* || *office desk* | | *snacks* ||
> |-|-|-|-|-| -| -| -| -|
> || **mIoU** | **mAP**|**mIoU**|**mAP**|**mIoU**|**mAP**|**mIoU**|**mAP**|
> | LSeg[3]      |18.7|68.8|19.3|77.0|15.3|36.4|27.1|81.3|
> |OV-Seg[4]     |74.8|88.5|73.7|53.7|85.5|74.8|75.3|46.1|
> |ODISE[5]      |55.2|47.6|63.8|72.2|61.1|69.9|33.6|63.7|
> | FFD[2]       |18.0|74.5|22.6|79.8|20.2|42.0|34.0|85.5|
> |Sem(ODISE)[1] |24.0|38.9|50.0|75.1|69.7|86.2|29.6|61.6|
> |Sem(OV-Seg)[1]|80.8|92.2|71.1|89.1|95.6|98.6|75.7|90.6|
> |LERF[6]       |30.9|40.0|74.5|91.1|55.6|77.0|61.0|85.2|
> | **Ours**     |82.8|97.2|88.6|97.2|91.7|96.2|95.8|99.1|
>
>
> Table2. Results of core ablations on all scenes.
> | scene | *table* || *bench* || *sofa* | | *room* ||*blue sofa*||
> |-|-|-|-|-| -| -| -| -| -| -|
> || **mIoU** | **mAP**|**mIoU**|**mAP**|**mIoU**|**mAP**|**mIoU**|**mAP**|**mIoU**|**mAP**|
> | w/o RDA      |60.1|36.4|52.4|89.6|39.0|39.2|73.4|97.4|44.7|93.3|
> |w/o FDA       |56.7|76.9|56.7|81.9|47.5|76.3|45.5|78.9|52.2|83.4|
> |w/o re-balance|15.8|53.4|22.1|55.4|19.9|69.0|22.5|56.5|52.3|83.4|
> |**ours**      |88.8|96.5|89.3|96.3|74.0|91.6|92.8|98.9|82.8|97.7|
>
>
> | scene | *coverd desk* || *bed* || *lawn* | | *office desk* ||*snacks*||
> |-|-|-|-|-| -| -| -| -| -| -|
> || **mIoU** | **mAP**|**mIoU**|**mAP**|**mIoU**|**mAP**|**mIoU**|**mAP**|**mIoU**|**mAP**|
> | w/o RDA      |70.7|91.5|88.6|96.8|56.9|94.0|22.0|64.3|64.3|92.0|
> |w/o FDA       |66.9|88.8|61.3|80.1|64.0|89.3|66.3|83.0|65.1|88.3|
> |w/o re-balance|67.1|88.9|61.2|80.0|63.7|89.2|62.3|83.0|63.0|88.5|
> |**ours**      |88.6|97.2|89.5|96.7|88.2|97.3|91.7|96.2|95.8|99.1|
>
> 5. **Shared Volume.** This design follows Semantic-NeRF [1] and FFD [2], which show that using the same intermediate features for both the RGB branch and the feature branch can lead to better results in lifting 2D features to 3D.
>
> [1] In-Place Scene Labelling and Understanding with Implicit Scene Representation
>
> [2] Decomposing NeRF for Editing via Feature Field Distillation
>
> [3] Language-driven semantic segmentation
>
> [4] Open-vocabulary semantic segmentation with mask-adapted clip
>
> [5] Open vocabulary panoptic segmentation with text-to-image diffusion models.
>
> [6] LERF: Language Embedded Radiance Fields

---

> > ### Comment · Reviewer_1C3c · 2023-08-18
> > **Response to author rebuttal**
> >
> > Thank authors for providing more visual reusts and clarifications, some of my concerns are clearly solved.
> >
> >
> > However,
> >
> > 1. as to additional experiments on indoors scenes, I feel the methods struggle to produce reasonable segmentations compared to other scenes (probably less clutterred and richer texture).
> >
> > 2. I have also read all reviews, I agree with some opinions from Reviewer kCD4 that :
> > the 'open-set 3D segmentation' is somewhat over-claimed (I understand the motivation and efforts in making language-guided 3d segmentation as discussed with reviewes), probably languided-guided/weakly-supervised, or other terms could better fit the context.
> >
> > Strictly speaking, a semantic prior need to be encoded in the text prompt for segmentation, compared to general promots 'a photo of something' in the classification set-up. Without such guidance, segmentation is challenging but this is partly why reviewers kCD4 insists on saying the title is over-claimed.
> >
> > Therefore, I would strongly encourage the authors to re-consider the title and be more conservative in using 'open-set 3d segmentation'.
> >
> > I do think there are still values in the method by leveraging two typical unsupervise representation models to solve the segmentation tasks,   but the limited performance on indoor scenes, the importance of semantics-guidance from text prompts, as well as more rigorous re-consideration in the overall story of the paper (e.g., titles, targeted tasks) should be clearly discussed and modified within the paper.
> >
> >
> > Overall, I keep my rating as 5 considering the potential value of this work but may not increase to 6 considering above limitations and unsolved issues.

---

### Author Rebuttal · Authors · 2023-08-06

We are sincerely thankful to every reviewer for reading our research narrowly and giving us thoughtful feedback. We carefully respond to the main concerns raised by the reviewers as below.

## 1 Evaluation Datasets

Reviewers **t9cD** and **kCD4** shared that our collected dataset has a limited scale for evaluating open-vocabulary features. We would clarify that 3D open-vocabulary segmentation is a relatively new task with few relevant benchmarks available to the best of our knowledge (so the very recent and related LERF [6] also evaluates on its own collected dataset). Specifically, most existing 3D datasets are either unannotated or have very limited object/scene classes. To tackle this issue, we purposely collected images of a wide range of diverse objects such as electronic products, fruits, animals, human body parts, brands, instruments, food, plants, etc., aiming to offer a new baseline for benchmarking on 3D open-vocabulary segmentation. In fact, our dataset is much more challenging than those used in the recent NeRF-based segmentation methods [1, 2] (from NeurIPS 2022 and ICLR 2023), from both object number per scene and complexity of scene geometry. Nevertheless, we managed to conduct more evaluations on a few other datasets as described below.


## 2 Evaluations on Other Datasets and Settings

We additionally perform evaluations on the human body dataset, human head dataset, indoor datasets with low-quality images [4, 5], and a complex scene from LERF datasets [6]. We compare with the ***concurrent*** work LERF [6] qualitatively due to the lack of labels or the defective annotations as pointed out in [3]. We also perform experiments with different text prompts. All the newly conducted qualitative experiments are presented in the submitted rebuttal PDF.

**Human body and head.** As shown in Figure 1 in the rebuttal PDF, our method segments more precise parts than LERF. Specifically, LERF fails to segment the "head" in the human body and the "black T-shirt" in the human head. In contrast, our method can recognize and segment these parts correctly because our designed RDA loss addresses the ambiguity of the CLIP features effectively.

**Indoor scenes with low-quality images.** Figure 2 in the rebuttal PDF shows experiments on the indoor datasets [4, 5], where many images are unrealistically rendered with less photorealistic appearances (as indicated in [3]) and have limited spatial resolution (640 $\times$ 480 or 1024 $\times$ 768). Due to these data constraints, our method sometimes confuses labels with similar appearances. However, we can see that our method still outperforms LERF by a large margin by successfully segmenting more labels.

**Complex scene.** Figure 3 (left) in the rebuttal PDF shows the segmentation of one challenging sample from the LERF dataset [6], where the scene has complex geometry as well as many objects of varying sizes. It can be observed that LERF cannot segment most objects due to the ambiguities of CLIP features while our method can segment more objects correctly with more precise boundaries.

**Segmentation with different text prompts.**  Reviewer **1C3c** raised questions about the impact of the text prompts, thus we conduct experiments to segment scenes with different text prompts.  In the experiments, we replaced the original texts with different languages (e.g., Portuguese egg tart -> Pastel de Nata),  names (e.g., dressing doll -> Barbie), and actions (e.g., hand soap -> wash hand, black headphone -> listen to music). As Figure 4 in the rebuttal PDF shows, with the rephrased text prompts, our method can still segment the scenes reliably. The experiments are well aligned with the quantitative experiments as shown in Table 1 below.

Table 1. Results with different text prompts.
| scene | *lawn* |  | *bench* | |
|-|-|-|-|-|
|    | **mIoU** | **mAP**   | **mIoU**  | **mAP**   |
| original   | 88.2   | 97.3   | 89.3   | 96.3  |
| rephrased  | 89.3  | 97.2  | 88.4  | 96.6  |


## 3 Novelty

Reviewers **kCD4** and **L8we** expressed concerns about the novelty of our approach, particularly when compared to FFD [6]. We would like to emphasize that our contribution is significantly different from FFD and other prior studies in three major aspects:

1. Our method directly distills open-vocabulary knowledge from CLIP, instead of relying on a fine-tuned version like LSeg. This enables the exploitation of the original large-scale vocabulary in CLIP.

2. Our method exploits two complementary foundation models, including DINO for precise object boundaries and CLIP for open vocabulary. In contrast, FFD and other methods like N3F [7] use a single foundation model at a time.

3. Our approach introduces a novel pipeline for 3D open-vocabulary segmentation using NeRF and achieves better performance than previous methods without requiring any segmentation annotations. Specifically, we introduce the Selection Volume to generate pixel-aligned CLIP features, RDA to mitigate the ambiguities of the CLIP features, and FDA for precise object boundaries.

**We firmly believe that our method can significantly benefit the 3D vision community by showcasing the possibility of learning 3D open-vocabulary segmentation from unlabeled 2D images and text-image pairs.**



[1] NeRF-SOS: Any-View Self-supervised Object Segmentation on Complex Scenes

[2] Unsupervised Multi-View Object Segmentation Using Radiance Field Propagation

[3] Decomposing NeRF for Editing via Feature Field Distillation

[4] Hypersim: A Photorealistic Synthetic Dataset for Holistic Indoor Scene Understanding

[5] The Replica Dataset: A Digital Replica of Indoor Spaces

[6] LERF: Language Embedded Radiance Fields

[7] Neural Feature Fusion Fields (N3F): 3D Distillation of Self-Supervised 2D Image Representations

---

### Decision · Program_Chairs · 2023-09-21

**Decision:**

Accept (poster)

**Comment:**

This paper proposes to train a NeRF model by leveraging two self-supervised trained models CLIP and DINO, tackling the problem of 3D scene segmentation. It receives two weak accepts, one borderline accept, one borderline reject and one reject.
During discussion, Reviewer kCD4 still got concerns on the comparison with LERF and the experiments for"Open-Vocabulary".
Reviewers had in-depth discussion and the AC recommends acceptance for this paper.
The authors are urged to clarify reviewers' concerns in the revision.